# C-TPT: Calibrated Test-Time Prompt Tuning for Vision-Language Models via Text Feature Dispersion

**Hee Suk Yoon**[1][*]    **Eunseop Yoon**[1][*]    **Joshua Tian Jin Tee**[1]
**Mark Hasegawa-Johnson**[2]    **Yingzhen Li**[3]    **Chang D. Yoo**[1][†]
[1]Korea Advanced Institute of Science and Technology (KAIST)
[2]University of Illinois at Urbana-Champaign (UIUC)    [3]Imperial College London
{hskyoon,esyoon97,joshuateetj,cd_yoo}@kaist.ac.kr
jhasegaw@illinois.edu  yingzhen.li@imperial.ac.uk

## Abstract

In deep learning, test-time adaptation has gained attention as a method for model fine-tuning without the need for labeled data. A prime exemplification is the recently proposed test-time prompt tuning for large-scale vision-language models such as CLIP. Unfortunately, these prompts have been mainly developed to improve accuracy, overlooking the importance of calibration, which is a crucial aspect for quantifying prediction uncertainty. However, traditional calibration methods rely on substantial amounts of labeled data, making them impractical for test-time scenarios. To this end, this paper explores calibration during test-time prompt tuning by leveraging the inherent properties of CLIP. Through a series of observations, we find that the prompt choice significantly affects the calibration in CLIP, where the prompts leading to higher text feature dispersion result in better-calibrated predictions. Introducing the Average Text Feature Dispersion (ATFD), we establish its relationship with calibration error and present a novel method, Calibrated Test-time Prompt Tuning (C-TPT), for optimizing prompts during test-time with enhanced calibration. Through extensive experiments on different CLIP architectures and datasets, we show that C-TPT can effectively improve the calibration of test-time prompt tuning without needing labeled data. The code is publicly accessible at https://github.com/hee-suk-yoon/C-TPT.

## 1 Introduction

Pre-trained large-scale vision-language models, such as CLIP (Radford et al., 2021), have demonstrated the potential as foundation models by leveraging their zero-shot inference capabilities. These models are pre-trained on a massive dataset of image-text caption pairs and learn to associate the image and its corresponding text caption in a shared latent space, which allows the model to accurately classify newfound visual categories in a zero-shot manner based on carefully designed prompt templates. As hand-crafted prompts consisting of predefined vocabulary tokens (i.e., hard prompts) may not be optimal, significant attention is being directed towards prompt tuning, which treats the prompts as learnable vectors that can be optimized through gradient descent. Specifically, Test-time Prompt Tuning (TPT) (Manli et al., 2022) adaptively refines the prompt for an individual unlabeled image sample. In line with well-known test-time adaptation methods for classifications (Wang et al., 2021; Zhang et al., 2022), TPT aims to enhance the accuracy of CLIP models by minimizing the entropy in the prediction distribution as a self-supervision signal during test time.

However, a reduction in entropy leads the model to generate overconfident predictions, which is a characteristic often observed in models trained with cross-entropy loss (Guo et al., 2017; Mukhoti et al., 2020; Eom et al., 2024). This overconfidence is intrinsically linked to worsening the model's calibration—a property that evaluates the degree to which the predicted probabilities align with the

---

[*]Equal contribution
[†]Corresponding Author

true underlying probability distribution (Guo et al., 2017). For instance, if a perfectly calibrated classifier assigns a confidence of 0.8 to its predictions, it should be correct 80% of the time. This property is particularly crucial in real-world applications where knowing the prediction uncertainty can ensure the trustworthiness and safety of the model. Although CLIP is increasingly employed in decision-making applications, such as healthcare diagnostics (Wang et al., 2022; Zhang et al., 2023; Chen et al., 2023a; Liu et al., 2023) and autonomous vehicles (Dorbala et al., 2022; Gadre et al., 2022; Khandelwal et al., 2022; Bucker et al.), calibration has been largely overlooked in existing CLIP prompt tuning literature, which has primarily focused on enhancing the classification accuracy.

This paper addresses this critical yet under-explored challenge of calibrated prompt tuning in large-scale vision-language models. Specifically, in light of the recent success of test-time prompt tuning on enhancing accuracy without labeled data (Manli et al., 2022), we aim to accomplish *calibration during test-time prompt tuning* to mitigate the adverse scenario where the prompt optimization, although enhancing accuracy, results in poor calibration. This may seem infeasible since various calibration techniques employed in standard supervised training of neural networks require substantial amounts of labeled training data, which restricts their applicability in test-time prompt tuning scenarios for models like CLIP. Instead, to enable calibration without labeled data, we capitalize on the intrinsic structures and properties of CLIP. **In detail, our contributions can be summarized as follows:**

- Through a series of observations, this paper reveals that the calibration of CLIP models is significantly influenced by the prompt choice, with certain prompts offering superior calibration with the same prediction accuracy level. We identify that the critical difference between these prompts can be characterized by the distribution of the class-embedded text features, with a noticeable negative correlation between the dispersion of the text features and the calibration error.

- This paper defines the Average Text Feature Dispersion (ATFD) and quantitatively establishes its strong negative correlation with Expected Calibration Error (ECE), reinforcing our finding of the relationship between the text feature dispersion and the calibration error.

- This paper introduces the Calibrated Test-time Prompt Tuning (C-TPT), which is used to jointly optimize the prompt during test time to achieve better calibration by maximizing ATFD. Extensive experiments demonstrate that across various datasets and CLIP models, incorporating C-TPT allows improved test-time optimization of the prompts, resulting in better-calibrated predictions.

## 2 RELATED WORK

**Prompt Tuning for Vision-Language Models**  Pre-trained vision-language models like CLIP (Radford et al., 2021) excel in zero-shot inference by learning to associate images and captions created using prompt templates. Prompt tuning treats the prompt templates as learnable vectors that can be optimized by gradient descent. For instance, CoOp (Zhou et al., 2022a) tunes the prompt in CLIP using a dataset of labeled training samples to improve its classification accuracy. However, CoCoOp (Zhou et al., 2022b) identifies that CoOp struggles with generalizing to out-of-distribution data and recommends conditioning the prompt on input images. While effective, these methods require access to annotated training data, which limits the zero-shot utilization of pre-trained models. To tackle this challenge, recent research has introduced Test-time Prompt Tuning (TPT) (Manli et al., 2022), a method that enables adaptive prompt learning on the fly using just one test sample.

**Calibration of Neural Network**  Calibration of neural networks measures the extent to which its prediction aligns with the true probability distribution (Guo et al., 2017). Among various methods for achieving better calibration, **post-processing calibration strategies** (Guo et al., 2017; Platt, 1999; Zadrozny and Elkan, 2001; 2002; Vovk et al., 2005; Lei et al., 2018; Pakdaman Naeini et al., 2015a) have been employed to calibrate an already trained model using a held-out validation dataset. Notable examples include temperature scaling (Guo et al., 2017), Platt scaling (Platt, 1999), and conformal prediction (Vovk et al., 2005; Lei et al., 2018). Alternatively, **trainable calibration methods** leverage a hybrid objective during the training of a neural network by combining a primary training loss with an auxiliary calibration objective loss. In this context, MMCE (Kumar et al., 2018), SB-ECE, S-AvUC (Karandikar et al., 2021), and ESD (Yoon et al., 2023a) offer differentiable calibration objective loss that can be used in the training process. Also, involving mixup augmentation (Zhang et al., 2018; Yoon et al., 2022) during training has been shown to improve the calibration of the model

(Thulasidasan et al., 2019; Wang et al., 2023). These methods often target calibration error reduction at the expense of a minor compromise in accuracy — typically less than 1 to 2% degradation.

However, the direct application of these widely established calibration methods shares a common limitation: *a reliance on substantial amounts of labeled data*, which restricts their applicability in few-shot or test-time prompt tuning scenarios in CLIP. To address these challenges, our work proposes a calibration technique tailored for prompt tuning in CLIP without labeled data.

## 3    BACKGROUND AND PROBLEM SETUP

**Zero-Shot Classification using Vision-Language Model (CLIP)**    The structure of CLIP consists of two separate encoders: visual encoder $\mathbf{F}_{\text{visual}}$ for transforming the image input into a visual feature vector and text encoder $\mathbf{F}_{\text{text}}$ for transforming textual inputs into text feature vectors. Suppose we have a single test image, $X_{\text{test}}$, belonging to class $Y_{\text{test}}$ for a classification problem involving $N$ distinct classes. In the fundamental zero-shot scenario with CLIP, we attach a manually designed prompt prefix (e.g., $\boldsymbol{p} =$"a photo of a") to each possible class $y_i$ of $Y \in \mathcal{Y} = \{y_1, y_2, \ldots, y_N\}$, generating class-related textual descriptions $[\mathbf{p}; y_i]$. Next, these class descriptions are fed into the text encoder to produce text features $\{\mathbf{t}_{[\mathbf{p};y_1]}, \mathbf{t}_{[\mathbf{p};y_2]}, \ldots, \mathbf{t}_{[\mathbf{p};y_N]}\}$, where $\mathbf{t}_{[\mathbf{p};y_i]} = \mathbf{F}_{\text{text}}([\mathbf{p}; y_i])$, and the test image is fed into the visual encoder to produce image feature $\mathbf{v} = \mathbf{F}_{\text{visual}}(X_{\text{test}})$. The image feature $\mathbf{v}$ is paired with each text feature $\mathbf{t}_{[\mathbf{p};y_i]}$ to determine a similarity score $s_i = \text{sim}(\mathbf{t}_{[\mathbf{p};y_i]} \cdot \mathbf{v})$, where $\text{sim}(\cdot)$ refers to the cosine similarity. The probability of predicting class $y_i$ for the test image $X_{\text{test}}$ can be described as $p(y_i|X_{\text{test}}) = \frac{\exp(\text{sim}(\mathbf{t}_{[\mathbf{p};y_i]} \cdot \mathbf{v})/\tau)}{\sum_{i=1}^{N} \exp(\text{sim}(\mathbf{t}_{[\mathbf{p};y_i]} \cdot \mathbf{v})/\tau)}$, where $\tau$ is the temperature of Softmax (set to 0.01 during inference). Then, the predicted class becomes $\hat{Y} = \arg\max_{y_i} p(y_i|X_{\text{test}})$ with its associated confidence as $\hat{P} = \max_{y_i} p(y_i|X_{\text{test}})$ .

**Prompt Tuning for CLIP**    Instead of hand-crafted prompts (i.e., hard prompts), prompt tuning has been adopted in various domains (Lester et al., 2021; Jia et al., 2022; Yoon et al., 2023b;c; Koo et al., 2024), which treat prompts as trainable embeddings, allowing optimization by gradient descent to maximize performance. Specifically, prior research on prompt tuning for CLIP (Zhou et al., 2022a;b; Chen et al., 2023b; Hantao Yao, 2023) tunes the prompt $\mathbf{p} \in \mathbb{R}^{L \times D}$ in the text embedding space, where $L$ represents the number of tokens and $D$ is the embedding size, using 16 samples per class from ImageNet dataset. These studies have demonstrated that the learned prompt can be generalized to different datasets. Recently, Test-time Prompt Tuning (TPT) (Manli et al., 2022) has emerged as an alternative approach that enables prompt optimization without requiring labeled data, achieving performance on par with traditional prompt tuning methods that rely on labeled training data.

**Calibration Error and Metric**    Consider an input $X$ with its corresponding label $Y$. For the input $X$ to a classifier, $\hat{Y}$ is the class prediction and $\hat{P}$ is its associated confidence. Formally, we define a classifier as perfectly calibrated when

$$\mathbb{P}(\hat{Y} = Y | \hat{P} = p) = p, \ \ \forall p \in [0, 1]. \tag{1}$$

In essence, this means that if a classifier assigns a confidence of $p$ to its predictions, then ideally, they should be correct $p$ proportion of the time. This property of calibration is orthogonal to accuracy, where the property of a classifier's accuracy cannot guarantee its calibration and vice versa. Mathematically, calibration can be quantified using metrics such as the Expected Calibration Error (ECE) (Pakdaman Naeini et al., 2015b). ECE is a binning-based metric that quantifies the calibration of a model's predicted probabilities. It works by dividing the confidence of predictions into equally sized bins and then measuring the discrepancy between the predicted and true probabilities within each bin. Given a set of images, their corresponding ground-truth class labels, and model predictions, the ECE can be computed as follows:

$$\text{ECE} = \sum_{k=1}^{K} \frac{|B_k|}{m} |\text{acc}(B_k) - \text{conf}(B_k)| , \tag{2}$$

where $K$ is the total number of bins, $B_k$ represents the set of images, $|B_k|$ denotes the number of images, $\text{acc}(B_k)$ is the accuracy of the predictions, and $\text{conf}(B_k)$ is the average prediction confidence, all respectively associated with bin $k$. A lower ECE value indicates better calibration, as the predicted probabilities are more closely aligned with the true probabilities of the prediction being correct.

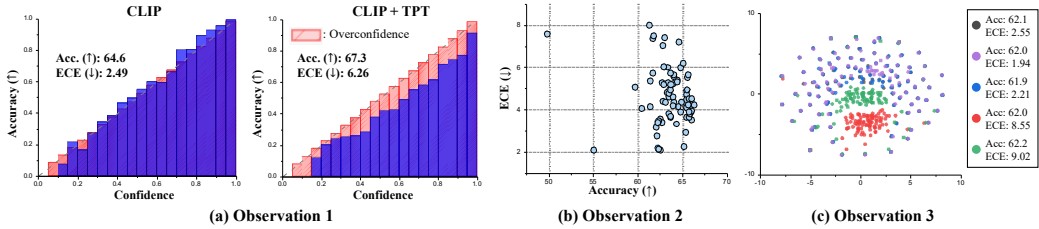

Figure 1: **Observations.** (The plots are based on the CLIP-ViT-B/16 on the StanfordCars dataset.) (a) *Observation 1* shows the Reliability Diagrams (Guo et al., 2017) of the prediction made with the hard prompt template ('an example of {class}') (*left*) and after applying TPT (*right*). The diagrams highlight the negative impact of TPT on calibration due to overconfident predictions. (b) *Observation 2* demonstrates the varying calibration error (i.e., ECE), although similar accuracy, plotted using 80 different hard prompt templates. (c) *Observation 3* features a t-SNE visualization of text feature clustering patterns of different prompts with similar accuracy but different ECE, suggesting that text feature dispersion has a strong relationship with the calibration error of CLIP.

## 4 REVISITING THE CALIBRATION OF CLIP MODELS

In this section, we introduce a series of observations revealing the intrinsic properties of CLIP which allows us to propose a calibration technique tailored for prompt tuning in CLIP without labeled data.

**Observation 1: Test-Time Prompt Tuning (TPT) Increases the Calibration Error.** Previous literature on *"test-time adaptation"* tries to improve accuracy by adapting the model to increase its prediction confidence on unlabeled target data (Mummadi et al., 2021). One of the mechanisms to achieve this is by employing an entropy minimization loss (Wang et al., 2021; Zhang et al., 2022) that uses maximization of the prediction confidence as a self-supervision signal during test-time adaptation. However, models trained with such loss are prone to overconfidence, which is one of the direct causes of calibration error (Guo et al., 2017; Mukhoti et al., 2020). We observe a parallel trend with Test-time prompt tuning (TPT) (Manli et al., 2022), which adopts a similar objective of entropy minimization during test-time to improve the classification accuracy of zero-shot inference using CLIP. As depicted in Figure 1-(a), *while applying TPT successfully enhances the accuracy over hard prompts, there is a corresponding trend of increase in calibration error.*

**Observation 2: Prompt Sensitivity in Calibration of CLIP Models.** The following observation begins by comparing the accuracy and calibration error (i.e., ECE) of various hard prompts to better comprehend the impact of the 'prompt' on the performance of CLIP. Figure 1-(b) shows the accuracy and ECE of zero-shot inference of CLIP using 80 different hard prompt templates[1]. Although previous literature suggests that CLIP models are generally well-calibrated (Minderer et al., 2021; Galil et al.), we discover that different prompts can yield notable variation in calibration performance while exhibiting similar accuracy (additional details in Appendix A.4). *This observation signifies that certain prompts could have better-calibrated predictions with similar accuracy levels, and thus, it is possible to guide the prompt optimization process toward better calibration.*

**Observation 3: Well-calibrated Prompts have High Text-Feature Dispersion.** Following the results from Observation 2, we further investigated why particular prompts give better-calibrated predictions than others, even when the accuracy level is similar. Recognizing that the image feature does not change for different prompts during classification using CLIP, we directed our attention toward the class-embedded text features. We visualized the t-SNE (van der Maaten, 2014) of text features from prompts with comparable accuracy, as depicted in Figure 1-(c), and identified a distinct pattern; poorly calibrated prompts typically led text features to cluster cohesively. In contrast, well-calibrated prompts manifested a wider dispersion in the text feature space. *This insight suggests that promoting text feature dispersion could be a guiding regularizer for test-time optimization, potentially mitigating our observed calibration issues in Observation 1.*

---

[1]The full list of prompt templates used can be found in Appendix A.3

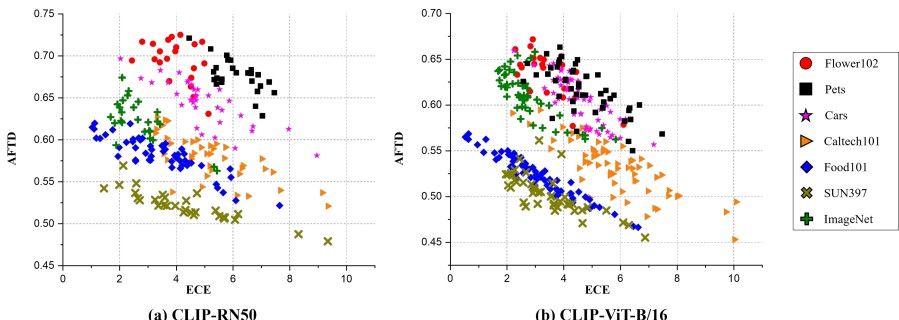

Figure 2: **Plot illustrating the correlation between ECE and ATFD** for hard prompts that achieve accuracies within 3% of the highest accuracy observed for each dataset. A notable negative association is observed for CLIP-RN50 and CLIP-ViT-B/16 across different datasets, with Pearson correlation coefficients (Freedman et al., 2007) averaging -0.70 and -0.76, respectively.

## 5   C-TPT: CALIBRATED TEST-TIME PROMPT TUNING

Motivated by our observations in Section 4, we introduce the concept of Average Text Feature Dispersion (ATFD) and examine its relationship with the calibration of CLIP in Section 5.1. Finally, we introduce our proposed Calibrated Test-Time Prompt Tuning (C-TPT) in Section 5.2.

### 5.1   CORRELATION BETWEEN CALIBRATION AND TEXT FEATURE DISPERSION

Observation 3 highlighted that well-calibrated prompts exhibit a wider dispersion of class-embedded text features. To quantitatively capture this dispersion for a given prompt $\mathbf{p}$, we first compute the centroid of the text features associated with each class description (i.e., $\mathbf{t}_{[\mathbf{p};y_1]}, \mathbf{t}_{[\mathbf{p};y_2]}, \ldots, \mathbf{t}_{[\mathbf{p};y_N]}$),

$$\mathbf{t}_{\text{centroid}} = \frac{1}{N} \sum_{i=1}^{N} \mathbf{t}_{[\mathbf{p};y_i]}. \tag{3}$$

Following this, we evaluate the spread of the text features by determining the mean L2 distance between this centroid and each individual text feature. This measure, termed Average Text Feature Dispersion (ATFD), is defined as:

$$\text{ATFD}(\mathbf{t}_{[\mathbf{p};y_1]}, \mathbf{t}_{[\mathbf{p};y_2]}, \ldots, \mathbf{t}_{[\mathbf{p};y_N]}) = \frac{1}{N} \sum_{i=1}^{N} ||\mathbf{t}_{\text{centroid}} - \mathbf{t}_{[\mathbf{p};y_i]}||_2. \tag{4}$$

A smaller ATFD indicates that the text features are closely clustered in the feature space, whereas a larger ATFD suggests a more dispersed distribution of the text features. Across multiple datasets, we conducted an evaluation using various hard prompts (Appendix A.3), calculating their respective accuracy and ECE. Subsequently, to rigorously understand the interplay between ATFD and ECE, we collated the prompts that yield similar accuracy, allowing for a more focused comparison. Our comprehensive analysis showed a notable negative correlation between ATFD and ECE across different CLIP models and datasets, as illustrated in Figure 2. This quantitative analysis reinforces our finding where the prompts leading to enhanced calibration are associated with a more dispersed distribution of text features across the possible classes.

### 5.2   C-TPT: CALIBRATED TEST-TIME PROMPT TUNING

Using our findings of the strong correlation between calibration and the Average Text Feature Dispersion (ATFD), we introduce Calibrated Test-time Prompt Tuning (C-TPT), which can be used to jointly train the prompt with tuning methods devised for accuracy (Figure 3). C-TPT focuses on tuning the prompt during test time to enhance calibration by spreading the text description features for each possible class by maximizing ATFD. Thus, our objective is formulated as:

$$\mathbf{p}^* = \arg\min_{\mathbf{p}} [\mathcal{L}_{\text{TPT}} + \lambda \cdot \mathcal{L}_{\text{C-TPT}}(\mathbf{t}_{[\mathbf{p};y_1]}, \mathbf{t}_{[\mathbf{p};y_2]}, \ldots, \mathbf{t}_{[\mathbf{p};y_N]})], \tag{5}$$

where $\mathcal{L}_{\text{C-TPT}} = -\text{ATFD}$ and $\mathbf{t}_{[\mathbf{p};y_1]}, \mathbf{t}_{[\mathbf{p};y_2]}, \ldots, \mathbf{t}_{[\mathbf{p};y_N]}$ are the class-embedded text representations.

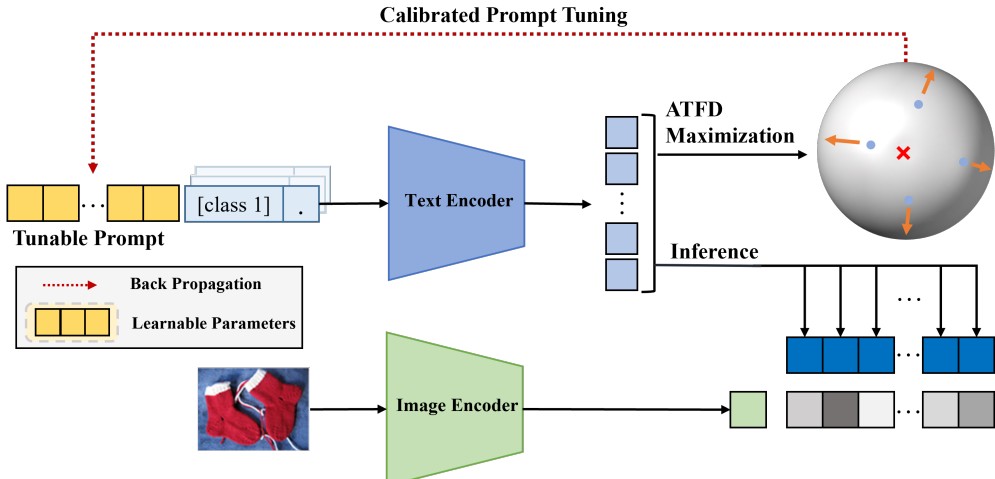

Figure 3: **Illustration of the Calibrated Test-time Prompt Tuning (C-TPT)** for zero-shot image classification using CLIP. C-TPT improves calibration by optimizing the prompt so that it maximizes the Average Text Feature Dispersion (ATFD) during test-time prompt tuning.

## 6 EXPERIMENTS

This section presents the benchmarks employed for evaluating our approach, discusses the implementation specifics, and the results of the experiments. In line with prior works on the prompt tuning of vision-language models (Zhou et al., 2022a;b; Chen et al., 2023b; Manli et al., 2022), our evaluation focuses on 1) multiple fine-grained classification and 2) natural distribution shift.

### 6.1 FINE-GRAINED CLASSIFICATION

**Dataset**   The evaluation is carried out across 11 diverse datasets for the fine-grained classification. These datasets include fine-grained classification of plants and animals (Flower102, OxfordPets), scene recognition (SUN397), texture identification (DTD), food categorization (Food101), transportation classification (StanfordCars, Aircraft), human action recognition (UCF101), satellite imagery analysis (EuroSAT), and general object classification (Caltech101, ImageNet). The details of each dataset are provided in Appendix A.5.

**Setup**   We showcase the effectiveness of C-TPT in reducing calibration error during test-time prompt tuning by incorporating it into the joint training with the previously proposed Test-time Prompt Tuning (TPT) (Manli et al., 2022). We initialize the prompt embeddings as the hard prompt 'a photo of a' (CLIP$_{\text{HardPrompt}}$) and optimize the corresponding embeddings using TPT (TPT$_{\text{HardPrompt}}$) or jointly using TPT and C-TPT (TPT$_{\text{HardPrompt}}$+C-TPT).

Moreover, we include an ensemble setting where we average the logits from 4 different hard-prompt initializations using {'a photo of a', 'a photo of the', 'a picture of a', 'a picture of the'} (CLIP$_{\text{Ensemble}}$). Similarly, we optimize using TPT (TPT$_{\text{Ensemble}}$), or jointly using TPT and C-TPT (TPT$_{\text{Ensemble}}$+C-TPT) on each of the hard-prompt initializations and average the resulting logits. Additional experiments with different prompt initializations can be found in Appendix A.9.

**Implementation Details**   We use CLIP-RN50 and CLIP-ViT-B/16 architectures. For all settings, we employ TPT (Manli et al., 2022) as the primary loss function to maximize accuracy while using C-TPT as an auxiliary objective to enhance calibration as in Eq. 5. We fix the $\lambda$ value at 50.0 for all cases [2], and in accordance with the previous test-time prompt tuning setup (Manli et al., 2022), we optimize the prompt for a single step using the AdamW (Loshchilov and Hutter) optimizer with a learning rate of 0.005. All other settings for training TPT are conducted following Manli et al. (2022). All experiments were performed on NVIDIA A100 80GB PCIe.

---

[2]The details on the choice of lambda is provided in Appendix A.6

Table 1: **Fine-Grained Classification**. We present the results of CLIP-RN50 and CLIP-ViT-B/16. For each setting, we report the **Acc.** (↑) and **ECE** (↓) of the initialization, after applying TPT, and after jointly employing TPT and our proposed C-TPT—the values highlighted in **bold** signify the best ECE achieved after test-time prompt tuning. Std. is reported in Appendix A.7.

| Method | | ImageNet | Caltech | Pets | Cars | Flower | Food101 | Aircraft | SUN397 | DTD | EuroSAT | UCF101 | Average |
|---|---|---|---|---|---|---|---|---|---|---|---|---|---|
| CLIP-RN50$_{\text{HardPrompt}}$ | Acc. | 58.1 | 85.8 | 83.8 | 55.7 | 61.0 | 74.0 | 15.6 | 58.6 | 40.0 | 23.7 | 58.4 | 55.9 |
| | ECE | 2.09 | 4.33 | 5.91 | 4.70 | 3.19 | 3.11 | 6.45 | 3.54 | 9.91 | 15.4 | 3.05 | 5.61 |
| +TPT$_{\text{HardPrompt}}$ | Acc. | 60.7 | 87.0 | 84.5 | 58.0 | 62.5 | 74.9 | 17.0 | 61.1 | 41.5 | 28.3 | 59.5 | 57.7 |
| | ECE | 11.4 | 5.04 | 3.65 | 3.76 | 13.4 | 5.25 | 16.1 | 9.24 | 25.7 | 22.5 | 12.4 | 11.7 |
| +TPT$_{\text{HardPrompt}}$+C-TPT | Acc. | 60.2 | 86.9 | 84.1 | 56.5 | 65.2 | 74.7 | 17.0 | 61.0 | 42.2 | 27.8 | 59.7 | 57.8 |
| | ECE | **3.01** | **2.07** | **2.77** | **1.94** | **4.14** | **1.86** | **10.7** | **2.93** | **19.8** | **15.1** | **3.83** | **6.20** |
| CLIP-RN50$_{\text{Ensemble}}$ | Acc. | 59.7 | 87.1 | 82.9 | 55.6 | 60.5 | 75.6 | 16.4 | 60.2 | 41.0 | 29.3 | 59.8 | 57.1 |
| | ECE | 5.15 | 6.43 | 6.46 | 7.34 | 5.02 | 5.04 | 3.92 | 6.19 | 4.54 | 7.70 | 3.55 | 5.58 |
| +TPT$_{\text{Ensemble}}$ | Acc. | 61.1 | 87.4 | 83.2 | 59.2 | 61.4 | 76.2 | 17.9 | 62.0 | 42.8 | 28.4 | 60.2 | 58.2 |
| | ECE | 11.2 | 4.29 | 4.79 | 3.08 | 14.1 | 5.27 | 14.6 | 7.68 | 22.2 | 18.9 | 11.1 | 10.7 |
| +TPT$_{\text{Ensemble}}$+C-TPT | Acc. | 61.2 | 87.4 | 84.0 | 57.3 | 65.3 | 76.0 | 17.5 | 62.1 | 43.1 | 29.4 | 60.7 | 58.5 |
| | ECE | **4.13** | **2.15** | **2.71** | **1.68** | **3.60** | **1.47** | **10.9** | **2.96** | **15.7** | **8.70** | **3.27** | **5.20** |
| CLIP-ViT-B/16$_{\text{HardPrompt}}$ | Acc. | 66.7 | 92.9 | 88.0 | 65.3 | 67.3 | 83.6 | 23.9 | 62.5 | 44.3 | 41.3 | 65.0 | 63.7 |
| | ECE | 2.12 | 5.50 | 4.37 | 4.25 | 3.00 | 2.39 | 5.11 | 2.53 | 8.50 | 7.40 | 3.59 | 4.43 |
| +TPT$_{\text{HardPrompt}}$ | Acc. | 69.0 | 93.8 | 87.1 | 66.3 | 69.0 | 84.7 | 23.4 | 65.5 | 46.7 | 42.4 | 67.3 | 65.0 |
| | ECE | 10.6 | 4.51 | 5.77 | 5.16 | 13.5 | 3.98 | 16.8 | 11.3 | 21.2 | 21.5 | 13.0 | 11.6 |
| +TPT$_{\text{HardPrompt}}$+C-TPT | Acc. | 68.5 | 93.6 | 88.2 | 65.8 | 69.8 | 83.7 | 24.0 | 64.8 | 46.0 | 43.2 | 65.7 | 64.8 |
| | ECE | **3.15** | **4.24** | **1.90** | **1.59** | **5.04** | **3.43** | **4.36** | **5.04** | **11.9** | **13.2** | **2.54** | **5.13** |
| CLIP-ViT-B/16$_{\text{Ensemble}}$ | Acc. | 68.2 | 93.4 | 86.3 | 65.4 | 65.7 | 85.2 | 23.5 | 64.0 | 45.6 | 43.0 | 66.1 | 64.2 |
| | ECE | 3.70 | 6.16 | 4.88 | 7.09 | 6.01 | 3.78 | 4.56 | 4.01 | 13.8 | 6.01 | 4.05 | 5.82 |
| +TPT$_{\text{Ensemble}}$ | Acc. | 69.6 | 94.1 | 86.1 | 67.1 | 67.6 | 85.1 | 24.4 | 66.5 | 47.2 | 44.0 | 68.5 | 65.5 |
| | ECE | 9.82 | 4.48 | 5.72 | 4.00 | 13.9 | 4.27 | 14.6 | 9.01 | 18.6 | 14.1 | 10.5 | 9.91 |
| +TPT$_{\text{Ensemble}}$+C-TPT | Acc. | 69.3 | 94.1 | 87.4 | 66.7 | 69.9 | 84.5 | 23.9 | 66.0 | 46.8 | 48.7 | 66.7 | 65.8 |
| | ECE | **4.48** | **3.14** | **1.54** | **1.84** | **5.77** | **2.38** | **6.40** | **3.09** | **13.7** | **5.49** | **3.04** | **4.62** |

**Results**  Prior to discussing our results, it is worth recapping the ideal outcome: *through the joint training with C-TPT, we desire to reduce the ECE relative to TPT while the accuracy drop is retained within a margin of 1%* (details in Appendix A.2 and A.8). Table 1 presents results for the fine-grained classification task. As noted in Observation 1 from Section 4, TPT improves the prediction accuracy but also results in a higher calibration error. However, when using both TPT and C-TPT, we observe better calibration across all settings without compromising the accuracy benefits of TPT.

Specifically, in the Hard Prompt initialization settings, the average ECE drops from 11.7 to 6.20 for CLIP-RN50 and from 11.6 to 5.13 for CLIP-ViT-B/16, representing 47% and 56% reduction in calibration error, respectively. In the Ensemble settings, the average ECE decreases from 10.7 to 5.20 for CLIP-RN50 and from 9.91 to 4.62 for CLIP-ViT-B/16, leading to a 52% and 53% reduction in calibration error, respectively. ***Furthermore, Appendix A.8 provides a visual representation illustrating the effectiveness of C-TPT.***

## 6.2 NATURAL DISTRIBUTION SHIFTS

**Dataset & Setup & Implementation Details**  For the natural distribution shift, we use ImageNet variants, including ImageNet-V2, ImageNet-A, ImageNet-R, and ImageNet-Sketch (details are in Appendix A.5). All the hyperparameters and configurations are set identically to Section 6.1 except for $\lambda$, which we set to 20.0.

**Results**  Table 2 shows the outcomes under natural distribution shifts. Similar to the results from Table 1, joint application of C-TPT improves the calibration error compared to utilizing TPT alone while still taking advantage of the accuracy increase of TPT. Specifically, for the Hard Prompt setting, there is a 28% and 52% drop of ECE for CLIP-RN50 and CLIP-ViT-B/16, respectively. For the Ensemble setting, there is a 14% and 24% drop in ECE, respectively.

Table 2: **Natural Distribution Shifts**. We report the **Acc.** (↑) and **ECE** (↓) of the initialization, after applying TPT, and after jointly employing TPT and our proposed C-TPT—the values highlighted in **bold** signify the best ECE achieved after test-time prompt tuning. Std. is reported in Appendix A.7.

| Method | | ImageNet-A | ImageNet-V2 | ImageNet-R | ImageNet-Sketch | **Average** |
|---|---|---|---|---|---|---|
| CLIP-RN50$_{HardPrompt}$ | Acc. | 21.7 | 51.4 | 56.0 | 33.3 | 40.6 |
| | ECE | 21.3 | 3.33 | 2.07 | 3.15 | 7.46 |
| +TPT$_{HardPrompt}$ | Acc. | 25.2 | 54.6 | 58.9 | 35.1 | 43.5 |
| | ECE | 31.0 | 13.1 | 9.18 | 13.7 | 16.7 |
| +TPT$_{HardPrompt}$+C-TPT | Acc. | 23.4 | 54.7 | 58.0 | 35.1 | 42.8 |
| | ECE | **25.4** | **8.58** | **4.57** | **9.70** | **12.1** |
| CLIP-RN50$_{Ensemble}$ | Acc. | 22.7 | 52.5 | 57.9 | 34.7 | 42.0 |
| | ECE | 17.0 | 2.68 | 5.64 | 10.9 | 9.06 |
| +TPT$_{Ensemble}$ | Acc. | 26.9 | 55.0 | 60.4 | 35.6 | 44.5 |
| | ECE | 29.1 | 12.7 | 7.50 | 14.0 | 15.8 |
| +TPT$_{Ensemble}$+C-TPT | Acc. | 25.6 | 54.8 | 59.7 | 35.7 | 44.0 |
| | ECE | **27.0** | **9.84** | **5.17** | **12.2** | **13.6** |
| CLIP-ViT-B/16$_{HardPrompt}$ | Acc. | 47.8 | 60.8 | 74.0 | 46.1 | 57.2 |
| | ECE | 8.61 | 3.01 | 3.58 | 4.95 | 5.04 |
| +TPT$_{HardPrompt}$ | Acc. | 52.6 | 63.0 | 76.7 | 47.5 | 59.9 |
| | ECE | 16.4 | 11.1 | 4.36 | 16.1 | 12.0 |
| +TPT$_{HardPrompt}$+C-TPT | Acc. | 51.6 | 62.7 | 76.0 | 47.9 | 59.6 |
| | ECE | **8.16** | **6.23** | **1.54** | **7.35** | **5.82** |
| CLIP-ViT-B/16$_{Ensemble}$ | Acc. | 50.9 | 62.0 | 74.5 | 46.0 | 58.4 |
| | ECE | 8.85 | 3.01 | 2.85 | 9.70 | 6.10 |
| +TPT$_{Ensemble}$ | Acc. | 54.2 | 63.9 | 78.2 | 48.5 | 61.2 |
| | ECE | 13.5 | 11.2 | 3.64 | 15.3 | 10.9 |
| +TPT$_{Ensemble}$+C-TPT | Acc. | 52.9 | 63.4 | 78.0 | 48.5 | 60.7 |
| | ECE | **10.9** | **8.38** | **1.40** | **12.6** | **8.32** |

# 7 ABLATION STUDY

## 7.1 COMPARISON WITH PREVIOUS CALIBRATION METHOD

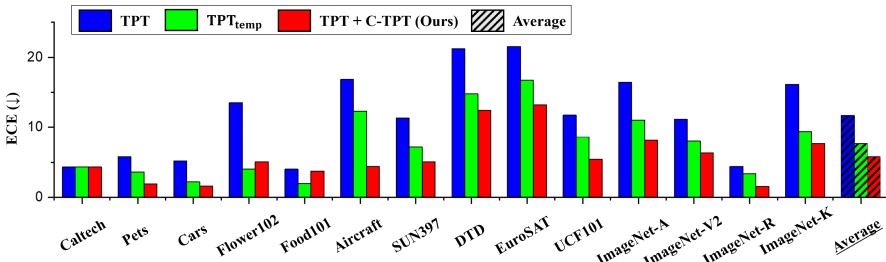

Figure 4: **Comparison of calibration error** between TPT, temperature-scaled TPT (TPT$_{temp}$), and the joint use of our proposed C-TPT (TPT+C-TPT). Results are based on CLIP-ViT-B/16.

Temperature scaling (Guo et al., 2017) is a widely used post-processing technique that calibrates predictions by adjusting the logits before softmax using a learned temperature value, which is trained on a separate held-out validation set. A recent study by LeVine et al. (2023) suggests that such a temperature can be generalized across various datasets and prompts in CLIP. In this section, we apply temperature scaling training with the TPT output logits of the entire labeled ImageNet test set. Subsequently, this learned temperature is employed for TPT on different datasets, including fine-grained classification and natural data shift datasets, to assess its generalization capabilities. As illustrated in Figure 4, the temperature-scaled approach (denoted as TPT$_{temp}$) does reduce calibration error compared to standard TPT. However, our proposed C-TPT exhibits better performance in most instances. Notably, C-TPT surpasses TPT$_{temp}$ in diminishing the average calibration error, even without leveraging any labeled training data.

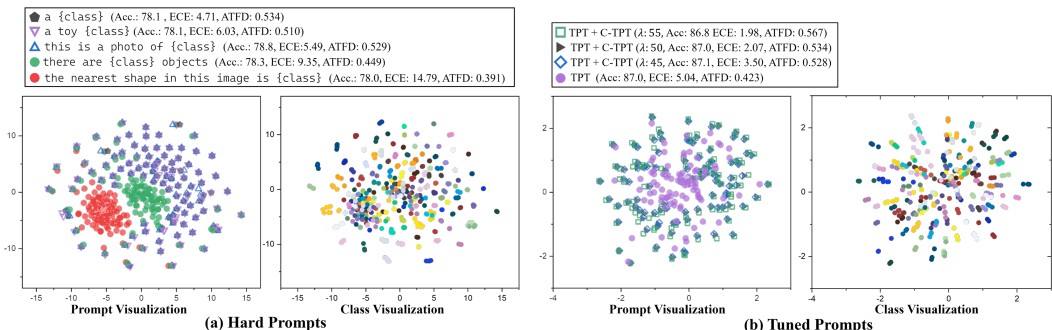

Figure 5: **t-SNE visualization of class-embedded textual representations** for (a) Hard Prompts and (b) Tuned Prompts, utilizing the CLIP-RN50 model on the Caltech101 dataset. In both cases, each unique color signifies a distinct prompt in the Prompt Visualization (*left*) and a distinct class in the Class Visualization (*right*). The legends belong to the Prompt Visualization (*left*) for both cases.

### 7.2 WHY DOES TEXT FEATURE DISPERSION INFLUENCE CALIBRATION?

To better understand the relationship between Average Text Feature Dispersion (ATFD) and Expected Calibration Error (ECE), we visualize the class-embedded textual representations using t-SNE (van der Maaten, 2014) plot of various hard prompts with similar accuracies but varying calibration (Figure 5-(a)). In Figure 5-(a) (*left*), in accordance with our findings, we can observe that the poorly-calibrated prompts ({*'there are {class} objects', 'the nearest shape in this image is'*}) are clustered together, showcasing a low ATFD. Conversely, the well-calibrated prompts ({*'a', 'a toy', 'this is a photo of'*}) are dispersed throughout the feature space, indicative of a high ATFD. Interestingly, for well-calibrated prompts, the features tend to cluster cohesively with respect to the class (Figure 5-(a) (*right*)), suggesting that the high dispersion in well-calibrated prompts is a result of the text features aligning closely to its respective class-label locations. On the contrary, in the case of poorly-calibrated prompts, the text features tend to cluster together regardless of the class labels. This pattern implies that certain intrinsic properties of these prompts such as their higher descriptiveness, could hinder the text features from clustering near their corresponding class labels, leading to poor calibration.

In Figure 5-(b) we replicate the previous analysis, but with a shift in focus to prompt tuning. Observations indicate that when we employ standard Test-time Prompt Tuning (TPT), the text descriptions appear to cluster with each other, much like the poorly calibrated hard prompts discussed earlier. However, as we apply our proposed C-TPT and progressively increase its influence during the optimization process by increasing the $\lambda$ value (i.e., 45, 50, 55), the text features become more dispersed (Figure 5-(b) (*left*)). Notably, analogous to the well-calibrated hard prompt scenario presented earlier, these features cluster in relation to their respective class labels (Figure 5-(b) (*right*)).

Within the hard prompt engineering, users may have control over calibration by selecting appropriate prompts that are not overly descriptive. However, this intuitiveness fades in prompt tuning, where prompts are sequences of embeddings rather than readable words. To address this, our proposed C-TPT can guide the optimization process for better calibration by increasing the text feature dispersion.

## 8 CONCLUSION

This paper presents the first comprehensive study of calibration in CLIP prompt tuning. A primary challenge in this domain is that the traditional calibration methods rely on extensive labeled data, making them unsuitable for prompt tuning scenarios. To navigate this constraint, we turn to the inherent properties of CLIP, uncovering the role of prompts in calibration. Despite comparable accuracy across prompts, calibration errors vary significantly due to the dispersion of class-embedded textual features. Well-calibrated prompts exhibit high feature dispersion near class labels, while poorly-calibrated prompts show low dispersion with clustering features. Building on this, we introduce Calibrated Test-time Prompt Tuning (C-TPT), which prioritizes the Average Text Feature Dispersion (ATFD) during tuning. Our experiments across various datasets demonstrate the effectiveness of C-TPT in refining the calibration of predictions without the dependence on labeled data.

## 9 ACKNOWLEDGEMENT

This work was supported by Institute of Information & communications Technology Planning & Evaluation (IITP) grant funded by the Korea government(MSIT) (No.2022-0-00184, Development and Study of AI Technologies to Inexpensively Conform to Evolving Policy on Ethics), and Institute of Information & communications Technology Planning & Evaluation (IITP) grant funded by the Korea government(MSIT) (No. 2022-0-00951, Development of Uncertainty-Aware Agents Learning by Asking Questions).

## 10 ETHICS STATEMENT

We confirm that our research adheres to the highest standards of ethical considerations. All work presented in this paper is original, and any external contributions or sources have been appropriately cited. Our study does not introduce new datasets, nor does it involve experiments utilizing demographic or identity characteristics.

## 11 REPRODUCIBILITY STATEMENT

To ensure the reproducibility of our work, we have detailed the comprehensive training procedures, hyperparameters, and experimental settings in Sections 6.1 and 6.2 of the paper. Moreover, we show the complete list of hard prompts used for analysis and experiments in Appendix A.3.

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

# A  APPENDIX

## A.1  BROADER IMPACT

This paper emphasizes the importance of calibration in vision-language foundation models, a critical yet often overlooked aspect in the drive for increasing accuracy in AI development. By achieving calibration without labeled data, a first in the field, it demonstrates how we can utilize the intrinsic properties of AI models for better calibration, setting a precedent for future research. The insights derived from this study could be particularly beneficial in enhancing the trustworthiness and safety of AI applications in scenarios where labeled data is scarce.

## A.2  LIMITATIONS

One limitation of regularization-based calibration methods is that there could be instances of accuracy degradation. Although the accuracy is not significantly affected based on our experimental results, there are some minor degradations in few cases. This slight reduction is in line with many trainable calibration methods (Kumar et al., 2018; Karandikar et al., 2021; Yoon et al., 2023a), which often target ECE reduction at the expense of minor compromise in accuracy— typically less than 1 to 2% degradation.

Another limitation worth noting is that since this paper leverages a test-time adaptation method, access to the training or validation sets for hyperparameter tuning is limited. Our approach involves the joint optimization of the accuracy-centric and calibration-centric loss, for which we introduce the balancing hyperparameter lambda ($\lambda$), as depicted in Eq. 5. In this study, we fix $\lambda$ to 50.0 for all test scenarios, as validated in Section A.6. While we acknowledge that this choice may not be optimal for every data sample, it represents a pragmatic initial step. Future research could explore strategies that dynamically adapt the balancing hyperparameter for each data sample, potentially enhancing model performance even further.

## A.3  LIST OF HARD PROMPTS USED FOR ANALYSIS

The plots and analyses depicted in Figure 1 and Figure 2 are derived from a curated set of 80 prompts originally developed by Radford et al. (2021). Theses prompts are listed as follows:

> **List of Hard Prompts**
>
> a drawing of the {class}, art of a {class}, itap of the {class}, a drawing of a {class}, a origami {class}, a photo of a nice {class}, a blurry photo of a {class}, a close-up photo of the {class}, a photo of a clean {class}, a photo of a weird {class}, a photo of a small {class}, a photo of the large {class}, a pixelated photo of the {class}, a embroidered {class}, a photo of the clean {class}, the origami {class}, the plushie {class}, a photo of a cool {class}, a sculpture of the {class}, a low resolution photo of the {class}, a bad photo of the {class}, a jpeg corrupted photo of a {class}, a rendition of the {class}, a photo of the cool {class}, a low resolution photo of a {class}, a cropped photo of the {class}, the plastic {class}, a sculpture of a {class}, a pixelated photo of a {class}, itap of a {class}, a doodle of a {class}, a sketch of a {class}, a plastic {class}, itap of my {class}, a close-up photo of a {class}, a bright photo of a {class}, art of the {class}, graffiti of the {class}, a tattoo of a {class}, a sketch of the {class}, a dark photo of a {class}, a tattoo of the {class}, a photo of the dirty {class}, a black and white photo of the {class}, a photo of a {class}, a painting of the {class}, a cropped photo of a {class}, a photo of a large {class}, a photo of the weird {class}, graffiti of a {class}, a painting of a {class}, a cartoon {class}, the cartoon {class}, a good photo of the {class}, a jpeg corrupted photo of the {class}, a bad photo of a {class}, a photo of the small {class}, a rendering of the {class}, a photo of a dirty {class}, a rendition of a {class}, a blurry photo of the {class}, the toy {class}, the embroidered {class}, a rendering of a {class}, a photo of a hard to see {class}, a dark photo of the {class}, a doodle of the {class}, a good photo of a {class}, a photo of the {class}, a photo of many {class}, a plushie {class}, a photo of the nice {class}, a bright photo of the {class}, a toy {class}, a photo of the hard to see {class}, a photo of one {class}, a photo of my {class}, a black and white photo of a {class}, a sketch of a {class}

## A.4 CALIBRATION OF CLIP MODELS

In Section 4, we illustrated the insight that emerges from our research, demonstrating that the choice of prompts can lead to varying calibration error, even when maintaining similar levels of accuracy. In Figure 6, we offer an additional visualization to further substantiate our finding of the pivotal role of prompt selection, not only in determining the accuracy but also the calibration error [3].

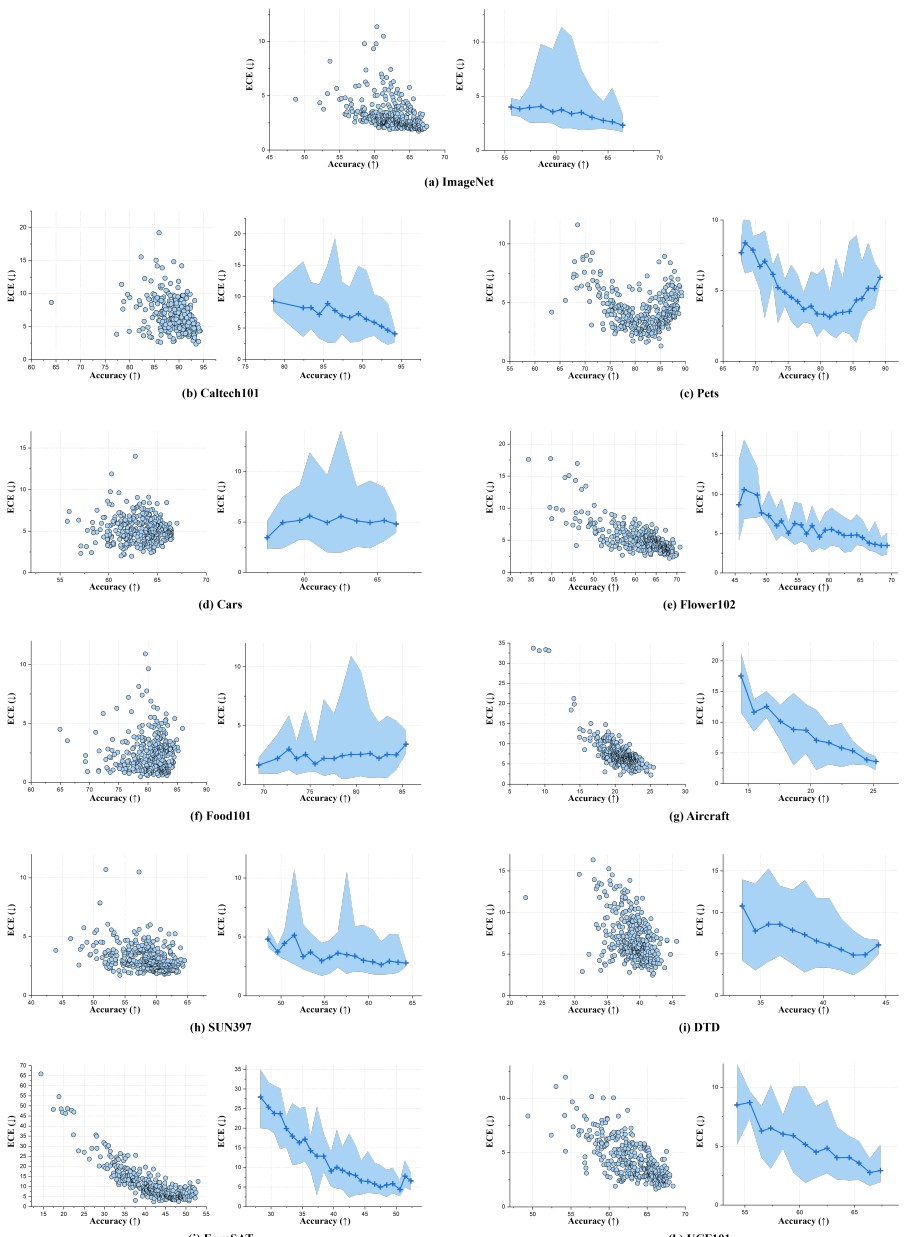

Figure 6: **Visualization of the variation in calibration error for different hard prompts** in CLIP-ViT-B/16 across 11 datasets. For each dataset, the *left* plot presents a scatter plot comparing the calibration error (ECE) and accuracy (Acc.) for a diverse set of prompts. In the *right* plot, we categorize the scatter points into bins, each encompassing a range of 1% accuracy. For each bin with more then 3 points, we plot the average ECE and Acc. as well as the bin's max and min ECE.

---

[3]In this section, we expand upon the initial set of 80 prompts specified in Section A.3, incorporating additional prompts provided by Allingham et al. (2023). This results in a diverse set of 320 hard prompts.

## A.5 DETAILS ON THE DATASET

In our experiments, we utilized the datasets adhering to CoOp (Zhou et al., 2022a). This encompasses 11 datasets designated for fine-grained classification and 4 ImageNet-derived datasets for evaluating natural distribution shifts. The details for each dataset, such as the number of classes, test-set size, and the respective tasks, can be found in Table 3.

Table 3: **The detailed statistics of datasets used in the experiments.**

| Dataset | # Classes | Test set size |
|---|---|---|
| ImageNet (Deng et al., 2009) | 1,000 | 50,000 |
| Caltech101 (Li et al., 2022) | 100 | 2,465 |
| OxfordPets (Parkhi et al., 2012) | 37 | 3,669 |
| StanfordCars (Krause et al., 2013) | 196 | 8,041 |
| Flowers102 (Nilsback and Zisserman, 2008) | 102 | 2,463 |
| Food101 (Bossard et al., 2014) | 101 | 30,300 |
| FGVCAircraft (Maji et al., 2013) | 100 | 3,333 |
| SUN397 (Xiao et al., 2010) | 397 | 19,850 |
| DTD (Cimpoi et al., 2014) | 47 | 1,692 |
| EuroSAT (Helber et al., 2018) | 10 | 8,100 |
| UCF101 (Soomro et al., 2012) | 101 | 3,783 |
| ImageNet-A (Hendrycks et al., 2021a) | 200 | 7,500 |
| ImageNetV2 (Recht et al., 2019) | 1,000 | 10,000 |
| ImageNet-R (Hendrycks et al., 2021b) | 200 | 30,000 |
| ImageNet-Sketch (Wang et al., 2019) | 1000 | 50,889 |

## A.6 INFORMATION ON THE CHOICE OF LAMBDA

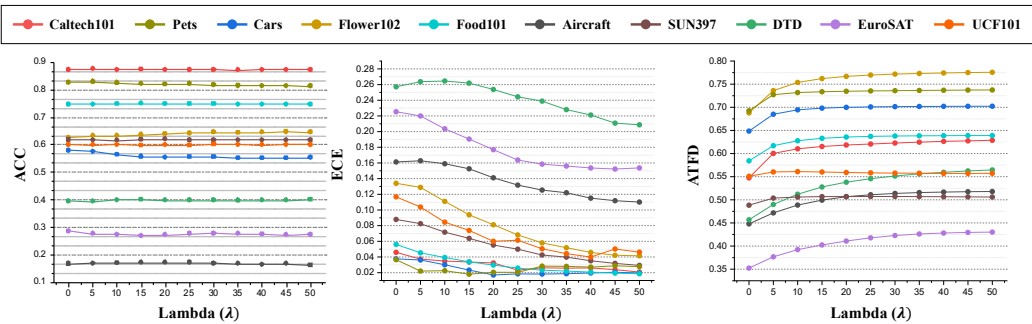

Figure 7: **The trend of Accuracy, ECE, and ATFD with respect to varying lambda** across the validation set of multiple datasets. Both the ECE and ATFD appear to saturate at equal to 50.0 across all datasets. While the accuracy is not significantly affected, there are instances of minor degradation of approximately 1%. The values are based on CLIP-RN50.

Our method employs a test-time prompt tuning strategy, which does not allow access to data for hyperparameter tuning in real-use cases. Nevertheless, our approach incorporates a lambda ($\lambda$) term as defined in Eq. 5. This term balances the accuracy-based objective and our proposed calibration objective, C-TPT. By analyzing the trend observed in the validation set of various datasets (depicted in Figure 7), both the accuracy and ATFD appear to saturate when $\lambda$ is at 50.0 across all datasets, suggesting that this value can be served as a generalizable hyperparameter regardless of the input. Consequently, in the experiments presented in our main paper, we set the $\lambda$ term to 50.0 during test-time optimization.

Additionally, an interesting observation worth highlighting from the plots presented in this section is their illustration of our key finding: the strong negative relationship between ECE and ATFD. As the ATFD increases according to increased $\lambda$, we can see a clear downward trend in ECE, thereby reinforcing our findings about their inverse relationship.

## A.7 RESULT: STANDARD DEVIATION

Table 4 and Table 5 presents the standard deviation of three random run with different seeds for the results of Table 1 and Table 2.

Table 4: **Standard Deviation of Fine-Grained Classification**.

| Method | | ImageNet | Caltech | Pets | Cars | Flower102 | Food101 | Aircraft | SUN397 | DTD | EuroSAT | UCF101 |
|---|---|---|---|---|---|---|---|---|---|---|---|---|
| CLIP-RN50_HardPrompt | - | - | - | - | - | - | - | - | - | - | - | - |
| +TPT_HardPrompt | Acc. | ±.03 | ±.24 | ±.14 | ±.16 | ±.12 | ±.16 | ±.32 | ±.07 | ±.35 | ±.13 | ±.24 |
| | ECE | ±.18 | ±.22 | ±.19 | ±.21 | ±.10 | ±.24 | ±.13 | ±.09 | ±.26 | ±.19 | ±.22 |
| +TPT_HardPrompt+C-TPT | Acc. | ±.04 | ±.10 | ±.35 | ±.23 | ±.18 | ±.26 | ±.22 | ±.10 | ±.20 | ±.13 | ±.17 |
| | ECE | ±.15 | ±.09 | ±.12 | ±.18 | ±.18 | ±.20 | ±.17 | ±.11 | ±.16 | ±.10 | ±.14 |
| CLIP-RN50_Ensemble | - | - | - | - | - | - | - | - | - | - | - | - |
| +TPT_Ensemble | Acc. | ±.11 | ±.22 | ±.20 | ±.26 | ±.32 | ±.24 | ±.25 | ±.26 | ±.13 | ±.21 | ±.19 |
| | ECE | ±.23 | ±.24 | ±.15 | ±.12 | ±.12 | ±.15 | ±.17 | ±.24 | ±.15 | ±.13 | ±.12 |
| +TPT_Ensemble+C-TPT | Acc. | ±.22 | ±.19 | ±.34 | ±.21 | ±.15 | ±.34 | ±.13 | ±.12 | ±.25 | ±.12 | ±.14 |
| | ECE | ±.17 | ±.14 | ±.20 | ±.22 | ±.19 | ±.11 | ±.17 | ±.16 | ±.22 | ±.15 | ±.13 |
| CLIP-ViT-B/16_HardPrompt | - | - | - | - | - | - | - | - | - | - | - | - |
| +TPT_HardPrompt | Acc. | ±.05 | ±.20 | ±.17 | ±.25 | ±.15 | ±.20 | ±.27 | ±.24 | ±.13 | ±.34 | ±.13 |
| | ECE | ±.19 | ±.23 | ±.17 | ±.24 | ±.17 | ±.12 | ±.10 | ±.21 | ±.13 | ±.18 | ±.11 |
| +TPT_HardPrompt+C-TPT | Acc. | ±.07 | ±.22 | ±.12 | ±.20 | ±.16 | ±.16 | ±.22 | ±.23 | ±.12 | ±.35 | ±.31 |
| | ECE | ±.12 | ±.12 | ±.15 | ±.19 | ±.18 | ±.24 | ±.19 | ±.10 | ±.24 | ±.08 | ±.20 |
| CLIP-ViT-B/16_Ensemble | - | - | - | - | - | - | - | - | - | - | - | - |
| +TPT_Ensemble | Acc. | ±.23 | ±.29 | ±.25 | ±.33 | ±.15 | ±.30 | ±.23 | ±.17 | ±.30 | ±.11 | ±.26 |
| | ECE | ±.11 | ±.11 | ±.24 | ±.21 | ±.10 | ±.22 | ±.08 | ±.19 | ±.12 | ±.23 | ±.15 |
| +TPT_Ensemble+C-TPT | Acc. | ±.32 | ±.26 | ±.18 | ±.20 | ±.12 | ±.22 | ±.24 | ±.11 | ±.10 | ±.20 | ±.29 |
| | ECE | ±.22 | ±.14 | ±.19 | ±.15 | ±.19 | ±.25 | ±.17 | ±.09 | ±.23 | ±.08 | ±.13 |

Table 5: **Standard Deviation of Natural Distribution Shifts**.

| Method | | ImageNet-A | ImageNet-V2 | ImageNet-R | ImageNet-Sketch |
|---|---|---|---|---|---|
| CLIP-RN50_HardPrompt | - | - | - | - | - |
| +TPT_HardPrompt | Acc. | ±.17 | ±.10 | ±.09 | ±.05 |
| | ECE | ±.19 | ±.20 | ±.16 | ±.13 |
| +TPT_HardPrompt+C-TPT | Acc. | ±.16 | ±.12 | ±.07 | ±.07 |
| | ECE | ±.18 | ±.11 | ±.13 | ±.09 |
| CLIP-RN50_Ensemble | - | - | - | - | - |
| +TPT_Ensemble | Acc. | ±.21 | ±.06 | ±.11 | ±.03 |
| | ECE | ±.13 | ±.18 | ±.13 | ±.16 |
| +TPT_Ensemble+C-TPT | Acc. | ±.17 | ±.07 | ±.10 | ±.05 |
| | ECE | ±.16 | ±.09 | ±.11 | ±.08 |
| CLIP-ViT-B/16_HardPrompt | - | - | - | - | - |
| +TPT_HardPrompt | Acc. | ±.19 | ±.11 | ±.07 | ±.09 |
| | ECE | ±.16 | ±.18 | ±.13 | ±.20 |
| +TPT_HardPrompt+C-TPT | Acc. | ±.09 | ±.09 | ±.06 | ±.08 |
| | ECE | ±.13 | ±.15 | ±.10 | ±.12 |
| CLIP-ViT-B/16_Ensemble | - | - | - | - | - |
| +TPT_Ensemble | Acc. | ±.21 | ±.12 | ±.08 | ±.04 |
| | ECE | ±.17 | ±.22 | ±.14 | ±.12 |
| +TPT_Ensemble+C-TPT | Acc. | ±.14 | ±.12 | ±.07 | ±.09 |
| | ECE | ±.18 | ±.10 | ±.13 | ±.09 |

## A.8  PARETO FRONT: VISUALIZING THE EFFECT OF C-TPT

Figure 8: Pareto Front of ECE vs. Accuracy.

In multi-objective optimization scenarios, there exists a trade-off between the two objectives. Similarly, calibration and accuracy are a trade-off in most regularization-based approaches (Kumar et al., 2018; Karandikar et al., 2021; Yoon et al., 2023a). This trade-off is often depicted through the Pareto front where each point on the front represents a different equilibrium between the conflicting objectives (Jin and Sendhoff, 2008; Lin et al., 2019; Navon et al., 2021). Figure 8 illustrates this concept by presenting a scatter plot comparison among various approaches: the hard prompt (CLIP), TPT, and our method (TPT+C-TPT). Figure 8 *left* shows the scatter plot where the points represent their corresponding accuracy and ECE using 80 different hard prompt initialization during the test-time prompt tuning. The plots show that our solution tends to form a Pareto front, rather than simply obtaining a prompt with lower ECE (i.e., better calibration) and lower accuracy. Figure 8 *right* focuses on the scatter plot for a specific hard prompt initialization ('a photo of a') under varying $\lambda$ term in Eq. 5. This visualization reveals how altering the $\lambda$ value causes the corresponding solutions to traverse along a Pareto front. Notably, this contrasts with the hard prompt, which clearly does not lie within the Pareto front.

## A.9 ADDITIONAL EXPERIMENTAL RESULTS: EXPLORING DIFFERENT INITIALIZATION

This section shows additional experimental results where the hard prompt is initialized differently. Table 6 shows the results when initializing the prompt with 'an example of', and Table 7 displays the results for the initialization 'a photo of the cool'. Moreover, we include the setting where the prompt is initialized with the supervised trained embeddings CoOp (Zhou et al., 2022a) and CoCoOp (Zhou et al., 2022b) in Table 8. For CoOp we utilize the officially published checkpoint and for CoCoOp we train on 16 images per class from half of the ImageNet's total classes with 4 learnable prompt embeddings.

Table 6: **Fine-Grained Classification**. TPT is initialized with 'an example of'. We report the **Acc.** (↑) and **ECE** (↓) of the initialization, after applying TPT, and after jointly employing TPT and our proposed C-TPT—the values highlighted in **bold** signify the best ECE achieved after test-time prompt tuning. Std. is reported in Appendix A.11.

| Method | | ImageNet | Caltech | Pets | Cars | Flower102 | Food101 | Aircraft | SUN397 | DTD | EuroSAT | UCF101 | Average |
|---|---|---|---|---|---|---|---|---|---|---|---|---|---|
| CLIP-RN50$_{HardPrompt}$ | Acc. | 55.1 | 80.3 | 75.7 | 55.8 | 58.1 | 75.2 | 16.1 | 56.2 | 41.1 | 25.5 | 56.3 | 54.1 |
| | ECE | 2.55 | 7.91 | 2.52 | 4.80 | 3.04 | 3.31 | 4.80 | 3.68 | 5.20 | 9.43 | 3.76 | 4.64 |
| +TPT$_{HardPrompt}$ | Acc. | 60.9 | 87.1 | 77.2 | 57.7 | 62.7 | 76.1 | 17.9 | 60.7 | 41.2 | 29.4 | 57.7 | 57.1 |
| | ECE | 8.51 | 5.12 | 6.98 | 5.52 | 12.2 | 4.83 | 15.2 | 8.19 | 20.2 | 11.1 | 15.3 | 10.3 |
| +TPT$_{HardPrompt}$+C-TPT | Acc. | 61.2 | 88.4 | 78.0 | 57.1 | 65.4 | 75.8 | 17.6 | 61.4 | 41.2 | 30.4 | 58.4 | 57.7 |
| | ECE | **3.85** | **2.89** | **2.72** | **2.05** | **2.97** | **1.90** | **7.16** | **4.84** | **15.6** | **7.69** | **6.99** | **5.33** |
| CLIP-ViT-B/16$_{HardPrompt}$ | Acc. | 65.2 | 90.9 | 82.5 | 64.6 | 64.7 | 83.9 | 22.3 | 61.4 | 42.4 | 38.8 | 64.8 | 62.0 |
| | ECE | 3.65 | 7.51 | 2.91 | 2.49 | 4.70 | 2.78 | 7.09 | 3.33 | 4.94 | 13.4 | 2.79 | 5.05 |
| +TPT$_{HardPrompt}$ | Acc. | 69.3 | 93.0 | 83.0 | 67.3 | 69.4 | 84.8 | 22.9 | 65.3 | 45.8 | 40.7 | 67.1 | 64.4 |
| | ECE | 8.72 | 2.91 | 7.30 | 6.26 | 12.2 | 5.05 | 16.2 | 7.94 | 20.5 | 20.8 | 11.6 | 10.9 |
| +TPT$_{HardPrompt}$+C-TPT | Acc. | 69.4 | 93.8 | 86.9 | 66.6 | 71.5 | 84.3 | 23.6 | 66.0 | 45.4 | 51.5 | 66.4 | 65.9 |
| | ECE | **4.04** | **1.62** | **2.89** | **1.75** | **4.49** | **1.36** | **9.05** | **3.54** | **15.5** | **5.18** | **3.87** | **4.85** |

Table 7: **Fine-Grained Classification**. TPT is initialized with 'a photo of the cool'. We report the **Acc.** (↑) and **ECE** (↓) of the initialization, after applying TPT, and after jointly employing TPT and our proposed C-TPT—the values highlighted in **bold** signify the best ECE achieved after test-time prompt tuning. Std. is reported in Appendix A.11.

| Method | | ImageNet | Caltech | Pets | Cars | Flower102 | Food101 | Aircraft | SUN397 | DTD | EuroSAT | UCF101 | Average |
|---|---|---|---|---|---|---|---|---|---|---|---|---|---|
| CLIP-RN50$_{HardPrompt}$ | Acc. | 56.9 | 80.9 | 79.8 | 56.9 | 57.7 | 73.0 | 16.1 | 56.5 | 39.6 | 21.9 | 56.3 | 54.1 |
| | ECE | 2.36 | 4.79 | 3.30 | 4.83 | 5.14 | 1.49 | 6.42 | 3.33 | 6.94 | 13.9 | 3.76 | 5.11 |
| +TPT$_{HardPrompt}$ | Acc. | 61.5 | 86.5 | 82.6 | 58.8 | 61.6 | 75.8 | 17.4 | 60.2 | 39.2 | 26.3 | 59.7 | 57.2 |
| | ECE | 12.6 | 6.02 | 7.31 | 4.49 | 17.0 | 7.93 | 17.5 | 11.4 | 24.8 | 15.7 | 14.4 | 12.7 |
| +TPT$_{HardPrompt}$+C-TPT | Acc. | 61.1 | 87.1 | 83.5 | 57.2 | 67.0 | 76.0 | 17.4 | 60.3 | 39.1 | 26.1 | 59.6 | 57.7 |
| | ECE | **8.74** | **2.85** | **1.75** | **1.65** | **6.34** | **3.70** | **13.5** | **8.28** | **18.0** | **11.2** | **8.82** | **7.71** |
| CLIP-ViT-B/16$_{HardPrompt}$ | Acc. | 64.7 | 88.1 | 86.2 | 66.5 | 64.5 | 81.4 | 22.7 | 62.4 | 38.4 | 34.6 | 67.6 | 61.6 |
| | ECE | 2.32 | 4.37 | 4.25 | 4.59 | 4.59 | 1.10 | 6.11 | 2.83 | 7.43 | 14.1 | 2.65 | 4.94 |
| +TPT$_{HardPrompt}$ | Acc. | 69.2 | 91.5 | 87.4 | 67.2 | 67.9 | 84.9 | 24.5 | 65.9 | 45.5 | 43.3 | 66.4 | 65.0 |
| | ECE | 12.2 | 3.11 | 6.34 | 6.36 | 14.6 | 5.74 | 19.2 | 13.3 | 20.0 | 18.2 | 14.1 | 12.1 |
| +TPT$_{HardPrompt}$+C-TPT | Acc. | 69.1 | 91.7 | 88.8 | 66.9 | 69.6 | 84.1 | 24.7 | 65.5 | 46.3 | 43.0 | 67.0 | 65.2 |
| | ECE | **7.27** | **1.89** | **1.59** | **1.64** | **10.6** | **2.43** | **10.5** | **10.7** | **18.0** | **8.73** | **7.42** | **7.34** |

Table 8: **Fine-Grained Classification**. TPT is initialized with the embeddings from CoOp and CoCoOp. We report the **Acc.** (↑) and **ECE** (↓) of the initialization, after applying TPT, and after jointly employing TPT and our proposed C-TPT—the values highlighted in **bold** signify the best ECE achieved after test-time prompt tuning. Std. is reported in Appendix A.11.

| Method | | ImageNet | Caltech | Pets | Cars | Flower102 | Food101 | Aircraft | SUN397 | DTD | EuroSAT | UCF101 | **Average** |
|---|---|---|---|---|---|---|---|---|---|---|---|---|---|
| CLIP-RN50$_{CoOp}$ | Acc. | 63.5 | 86.6 | 87.1 | 55.4 | 61.4 | 75.6 | 15.1 | 58.2 | 37.3 | 26.2 | 59.0 | 56.9 |
| | ECE | 1.97 | 3.12 | 6.65 | 7.17 | 4.59 | 2.56 | 6.20 | 1.99 | 10.4 | 12.2 | 3.92 | 5.52 |
| +TPT$_{CoOp}$ | Acc. | 65.5 | 87.3 | 87.6 | 57.7 | 61.5 | 76.4 | 15.8 | 60.0 | 38.2 | 27.1 | 60.8 | 58.0 |
| | ECE | 17.1 | 7.71 | 4.86 | 5.63 | 20.1 | 11.0 | 23.0 | 21.1 | 35.7 | 32.1 | 21.4 | 18.2 |
| +TPT$_{CoOp}$+C-TPT | Acc. | 64.2 | 87.6 | 87.4 | 57.0 | 62.2 | 76.0 | 15.4 | 59.4 | 37.7 | 26.3 | 60.1 | 57.6 |
| | ECE | **7.53** | **3.26** | **3.14** | **2.57** | **7.75** | **4.39** | **14.4** | **11.2** | **27.1** | **19.7** | **11.0** | **10.2** |
| CLIP-RN50$_{CoCoOp}$ | Acc. | 61.5 | 88.6 | 87.4 | 54.7 | 64.0 | 75.3 | 14.0 | 60.8 | 39.2 | 29.1 | 61.5 | 57.8 |
| | ECE | 2.63 | 4.24 | 7.39 | 6.14 | 3.86 | 2.52 | 4.15 | 3.79 | 7.25 | 5.24 | 2.63 | 4.53 |
| +TPT$_{CoCoOp}$ | Acc. | 62.8 | 89.1 | 88.0 | 56.5 | 64.1 | 75.9 | 15.0 | 62.0 | 39.4 | 29.2 | 60.0 | 58.4 |
| | ECE | 3.71 | **1.59** | 4.39 | **2.30** | 3.85 | 2.67 | 7.31 | 4.50 | 15.0 | 6.31 | 7.87 | 5.41 |
| +TPT$_{CoCoOp}$+C-TPT | Acc. | 62.0 | 88.7 | 87.6 | 55.5 | 64.1 | 75.7 | 14.3 | 61.3 | 39.6 | 28.8 | 59.1 | 57.9 |
| | ECE | **1.55** | 2.90 | 6.29 | 4.42 | **3.31** | **1.38** | **5.70** | **2.14** | **9.82** | **6.14** | **4.08** | **4.33** |
| CLIP-ViT-B/16$_{CoOp}$ | Acc. | 71.6 | 93.6 | 89.2 | 63.1 | 67.4 | 83.2 | 18.0 | 63.7 | 43.1 | 40.1 | 66.0 | 63.5 |
| | ECE | 1.41 | 3.65 | 2.92 | 6.86 | 3.92 | 1.55 | 9.21 | 1.72 | 7.71 | 15.3 | 3.47 | 5.25 |
| +TPT$_{CoOp}$ | Acc. | 73.7 | 94.0 | 89.1 | 65.6 | 68.7 | 83.8 | 20.0 | 65.6 | 44.5 | 40.6 | 67.2 | 64.8 |
| | ECE | 15.2 | 3.65 | 7.40 | 6.63 | 19.9 | 9.66 | 29.6 | 20.8 | 34.8 | 31.3 | 19.9 | 18.1 |
| +TPT$_{CoOp}$+C-TPT | Acc. | 72.5 | 93.9 | 89.3 | 63.1 | 69.0 | 83.7 | 19.2 | 65.1 | 45.0 | 40.4 | 66.6 | 64.3 |
| | ECE | **6.80** | **1.66** | **2.12** | **2.45** | **10.2** | **4.49** | **21.5** | **11.8** | **21.0** | **13.2** | **12.0** | **9.75** |
| CLIP-ViT-B/16$_{CoCoOp}$ | Acc. | 67.8 | 91.0 | 88.3 | 64.9 | 68.4 | 84.1 | 24.2 | 63.0 | 44.6 | 44.1 | 67.0 | 64.3 |
| | ECE | 2.89 | 3.52 | 4.60 | 6.51 | 3.82 | 3.25 | 4.06 | 4.61 | 3.82 | 5.81 | 3.28 | 4.20 |
| +TPT$_{CoCoOp}$ | Acc. | 68.8 | 91.2 | 88.5 | 65.9 | 68.6 | 84.6 | 24.9 | 64.0 | 45.0 | 44.5 | 67.8 | 64.9 |
| | ECE | 2.33 | **2.74** | **2.22** | 5.22 | 4.70 | **1.94** | 6.13 | 3.16 | 6.91 | 9.03 | 3.47 | 4.35 |
| +TPT$_{CoCoOp}$+C-TPT | Acc. | 68.4 | 91.4 | 88.8 | 64.9 | 69.3 | 84.2 | 24.6 | 63.6 | 44.7 | 44.3 | 67.1 | 64.7 |
| | ECE | **1.72** | 3.45 | 3.76 | **5.09** | **3.13** | 2.66 | **4.80** | **2.96** | **4.18** | **5.79** | **2.91** | **3.68** |

## A.10 ADDITIONAL EXPERIMENTAL RESULTS: APPLICATION TO DIFFERENT TEST-TIME PROMPT TUNING METHODS

In this Section, we further show that C-TPT can also be applied to other test-time prompt tuning methods. Specifically, we apply C-TPT to PromptAlign (Hassan et al., 2023), which introduces a distribution alignment strategy for CLIP to improve test-time adaptation.

We initialize the prompt embeddings as the hard prompt 'a photo of a' and optimize the corresponding embeddings using PromptAlign (PromptAlign$_{\text{HardPrompt}}$) or jointly using PromptAlign and C-TPT (PromptAlign$_{\text{HardPrompt}}$ + C-TPT). The experimental settings for applying PromptAlign follow its original settings from Hassan et al. (2023). As with our previous experiments, we fixed the $\lambda$ value to 50.0 for Fine-Grained Classification and 20.0 for Natural Distribution Shifts. Table 9 presents the results for the fine-grained classification task and Table 10 shows the outcomes under natural distribution shift task.

The results show that C-TPT can effectively reduce the calibration error while still taking advantage of the accuracy increase of PromptAlign. This signifies that other test-time prompt tuning methods can benefit from C-TPT to decrease the calibration error.

Table 9: **Fine-Grained Classification**. PromptAlign is initialized with 'a photo of a'. We report the **Acc.** (↑) and **ECE** (↓) of the initialization, after applying PromptAlign, and after jointly employing PromptAlign and our proposed C-TPT—the values highlighted in **bold** signify the best ECE achieved after test-time prompt tuning. Std. is reported in Appendix A.11.

| Method | | ImageNet | Caltech | Pets | Cars | Flower102 | Food101 | Aircraft | SUN397 | DTD | EuroSAT | UCF101 | Average |
|---|---|---|---|---|---|---|---|---|---|---|---|---|---|
| CLIP-ViT-B/16$_{\text{HardPrompt}}$ | Acc. | 65.2 | 90.9 | 82.5 | 64.6 | 64.7 | 83.9 | 22.3 | 61.4 | 42.4 | 38.8 | 64.8 | 62.0 |
| | ECE | 3.65 | 7.51 | 2.91 | 2.49 | 4.70 | 2.78 | 7.09 | 3.33 | 4.94 | 13.4 | 2.79 | 5.05 |
| +PromptAlign$_{\text{HardPrompt}}$ | Acc. | 72.1 | 94.1 | 90.5 | 68.0 | 72.1 | 87.6 | 25.5 | 68.1 | 47.9 | 44.8 | 69.8 | 67.3 |
| | ECE | 7.69 | 2.30 | 2.86 | 1.98 | 11.2 | 3.04 | 8.30 | 8.39 | 25.6 | 24.7 | 12.1 | 9.83 |
| +PromptAlign$_{\text{HardPrompt}}$+C-TPT | Acc. | 72.2 | 94.0 | 90.6 | 67.8 | 72.1 | 87.5 | 25.3 | 67.8 | 47.7 | 45.9 | 69.8 | 67.3 |
| | ECE | **5.87** | **2.20** | **2.09** | **1.79** | **9.26** | **2.25** | **6.57** | **6.29** | **22.1** | **21.8** | **9.95** | **8.19** |

Table 10: **Natural Distribution Shifts**. We report the **Acc.** (↑) and **ECE** (↓) of the initialization, after applying PromptAlign, and after jointly employing PromptAlign and our proposed C-TPT—the values highlighted in **bold** signify the best ECE achieved after test-time prompt tuning. Std. is reported in Appendix A.11.

| Method | | ImageNet-A | ImageNet-V2 | ImageNet-R | ImageNet-Sketch | **Average** |
|---|---|---|---|---|---|---|
| CLIP-ViT-B/16$_{\text{HardPrompt}}$ | Acc. | 47.8 | 60.8 | 74.0 | 46.1 | 57.2 |
| | ECE | 8.61 | 3.01 | 3.78 | 4.95 | 5.09 |
| +PromptAlign$_{\text{HardPrompt}}$ | Acc. | 55.8 | 65.3 | 78.5 | 50.3 | 62.5 |
| | ECE | 16.1 | 10.8 | 4.70 | 16.8 | 12.1 |
| +PromptAlign$_{\text{HardPrompt}}$+C-TPT | Acc. | 55.6 | 65.1 | 78.6 | 50.2 | 62.4 |
| | ECE | **13.1** | **7.10** | **2.98** | **12.8** | **9.00** |

## A.11 ADDITIONAL EXPERIMENTAL RESULTS: STANDARD DEVIATION

The followings are the standard deviation of the experiments done in Appendix A.9.

Table 11: **Standard Deviation of the initialization using 'an example of'.**

| Method | | ImgNet | Caltech | Pets | Cars | Flower | Food101 | Aircraft | SUN397 | DTD | EuroSAT | UCF101 |
|---|---|---|---|---|---|---|---|---|---|---|---|---|
| CLIP-RN50$_{HardPrompt}$ | - | - | - | - | - | - | - | - | - | - | - | - |
| +TPT$_{HardPrompt}$ | Acc. | ±.09 | ±.16 | ±.25 | ±.20 | ±.18 | ±.23 | ±.32 | ±.14 | ±.24 | ±.20 | ±.17 |
| | ECE | ±.13 | ±.13 | ±.14 | ±.13 | ±.22 | ±.13 | ±.18 | ±.12 | ±.19 | ±.25 | ±.14 |
| +TPT$_{HardPrompt}$+C-TPT | Acc. | ±.10 | ±.20 | ±.11 | ±.24 | ±.16 | ±.16 | ±.21 | ±.22 | ±.22 | ±.13 | ±.16 |
| | ECE | ±.11 | ±.10 | ±.12 | ±.11 | ±.17 | ±.24 | ±.25 | ±.16 | ±.20 | ±.20 | ±.25 |
| CLIP-ViT-B/16$_{HardPrompt}$ | - | - | - | - | - | - | - | - | - | - | - | - |
| +TPT$_{HardPrompt}$ | Acc. | ±.22 | ±.17 | ±.21 | ±.22 | ±.24 | ±.15 | ±.21 | ±.25 | ±.15 | ±.14 | ±.17 |
| | ECE | ±.20 | ±.12 | ±.25 | ±.20 | ±.15 | ±.13 | ±.14 | ±.21 | ±.20 | ±.12 | ±.18 |
| +TPT$_{HardPrompt}$+C-TPT | Acc. | ±.20 | ±.16 | ±.17 | ±.11 | ±.15 | ±.23 | ±.20 | ±.22 | ±.23 | ±.17 | ±.13 |
| | ECE | ±.21 | ±.15 | ±.11 | ±.21 | ±.18 | ±.17 | ±.11 | ±.17 | ±.16 | ±.18 | ±.24 |

Table 12: **Standard Deviation of the initialization using 'a photo of the cool'.**

| Method | | ImgNet | Caltech | Pets | Cars | Flower | Food101 | Aircraft | SUN397 | DTD | EuroSAT | UCF101 |
|---|---|---|---|---|---|---|---|---|---|---|---|---|
| CLIP-RN50$_{HardPrompt}$ | - | - | - | - | - | - | - | - | - | - | - | - |
| +TPT$_{HardPrompt}$ | Acc. | ±.10 | ±.19 | ±.13 | ±.13 | ±.25 | ±.29 | ±.21 | ±.28 | ±.24 | ±.15 | ±.19 |
| | ECE | ±.16 | ±.13 | ±.18 | ±.19 | ±.24 | ±.16 | ±.13 | ±.31 | ±.22 | ±.23 | ±.17 |
| +TPT$_{HardPrompt}$+C-TPT | Acc. | ±.11 | ±.14 | ±.26 | ±.28 | ±.16 | ±.14 | ±.25 | ±.15 | ±.18 | ±.29 | ±.25 |
| | ECE | ±.19 | ±.14 | ±.24 | ±.29 | ±.28 | ±.20 | ±.12 | ±.25 | ±.14 | ±.19 | ±.27 |
| CLIP-ViT-B/16$_{HardPrompt}$ | - | - | - | - | - | - | - | - | - | - | - | - |
| +TPT$_{HardPrompt}$ | Acc. | ±.08 | ±.28 | ±.27 | ±.21 | ±.15 | ±.26 | ±.29 | ±.32 | ±.24 | ±.21 | ±.21 |
| | ECE | ±.13 | ±.18 | ±.24 | ±.18 | ±.13 | ±.12 | ±.30 | ±.16 | ±.25 | ±.16 | ±.14 |
| +TPT$_{HardPrompt}$+C-TPT | Acc. | ±.11 | ±.19 | ±.18 | ±.13 | ±.26 | ±.30 | ±.19 | ±.30 | ±.28 | ±.23 | ±.23 |
| | ECE | ±.12 | ±.22 | ±.12 | ±.22 | ±.30 | ±.28 | ±.21 | ±.24 | ±.18 | ±.13 | ±.25 |

Table 13: **Standard Deviation of the initialization using CoOp and CoCoOp.**

| Method | | ImgNet | Caltech | Pets | Cars | Flower | Food101 | Aircraft | SUN397 | DTD | EuroSAT | UCF101 |
|---|---|---|---|---|---|---|---|---|---|---|---|---|
| CLIP-RN50$_{CoOp}$ | - | - | - | - | - | - | - | - | - | - | - | - |
| +TPT$_{CoOp}$ | Acc. | ±.11 | ±.22 | ±.20 | ±.26 | ±.32 | ±.24 | ±.25 | ±.26 | ±.13 | ±.21 | ±.19 |
| | ECE | ±.23 | ±.24 | ±.15 | ±.12 | ±.12 | ±.15 | ±.17 | ±.24 | ±.15 | ±.13 | ±.12 |
| +TPT$_{CoOp}$+C-TPT | Acc. | ±.22 | ±.19 | ±.34 | ±.21 | ±.15 | ±.34 | ±.13 | ±.12 | ±.25 | ±.12 | ±.14 |
| | ECE | ±.17 | ±.14 | ±.20 | ±.22 | ±.19 | ±.11 | ±.17 | ±.16 | ±.22 | ±.15 | ±.13 |
| CLIP-RN50$_{CoCoOp}$ | - | - | - | - | - | - | - | - | - | - | - | - |
| +TPT$_{CoCoOp}$ | Acc. | ±.24 | ±.32 | ±.33 | ±.27 | ±.33 | ±.35 | ±.28 | ±.16 | ±.31 | ±.10 | ±.20 |
| | ECE | ±.11 | ±.12 | ±.24 | ±.21 | ±.12 | ±.17 | ±.14 | ±.13 | ±.11 | ±.19 | ±.10 |
| +TPT$_{CoCoOp}$+C-TPT | Acc. | ±.34 | ±.17 | ±.34 | ±.16 | ±.32 | ±.24 | ±.10 | ±.34 | ±.20 | ±.30 | ±.16 |
| | ECE | ±.15 | ±.12 | ±.23 | ±.16 | ±.14 | ±.20 | ±.18 | ±.15 | ±.15 | ±.19 | ±.19 |
| CLIP-ViT-B/16$_{CoOp}$ | - | - | - | - | - | - | - | - | - | - | - | - |
| +TPT$_{CoOp}$ | Acc. | ±.23 | ±.29 | ±.25 | ±.33 | ±.15 | ±.30 | ±.23 | ±.17 | ±.30 | ±.11 | ±.26 |
| | ECE | ±.11 | ±.11 | ±.24 | ±.21 | ±.10 | ±.22 | ±.08 | ±.19 | ±.12 | ±.23 | ±.15 |
| +TPT$_{CoOp}$+C-TPT | Acc. | ±.32 | ±.26 | ±.18 | ±.20 | ±.12 | ±.22 | ±.24 | ±.11 | ±.10 | ±.20 | ±.29 |
| | ECE | ±.22 | ±.14 | ±.19 | ±.15 | ±.19 | ±.25 | ±.17 | ±.09 | ±.23 | ±.08 | ±.13 |
| CLIP-ViT-B/16$_{CoCoOp}$ | - | - | - | - | - | - | - | - | - | - | - | - |
| +TPT$_{CoCoOp}$ | Acc. | ±.22 | ±.29 | ±.31 | ±.12 | ±.17 | ±.15 | ±.20 | ±.30 | ±.33 | ±.17 | ±.33 |
| | ECE | ±.17 | ±.19 | ±.14 | ±.16 | ±.19 | ±.11 | ±.17 | ±.11 | ±.19 | ±.22 | ±.15 |
| +TPT$_{CoCoOp}$+C-TPT | Acc. | ±.21 | ±.13 | ±.25 | ±.21 | ±.25 | ±.19 | ±.24 | ±.13 | ±.22 | ±.22 | ±.23 |
| | ECE | ±.24 | ±.18 | ±.12 | ±.12 | ±.14 | ±.12 | ±.22 | ±.17 | ±.09 | ±.15 | ±.13 |

Table 14: **Standard Deviation of applying PromptAlign on Fine-grained Classification Task.**

| Method | | ImgNet | Caltech | Pets | Cars | Flower | Food101 | Aircraft | SUN397 | DTD | EuroSAT | UCF101 |
|---|---|---|---|---|---|---|---|---|---|---|---|---|
| CLIP-ViT-B/16$_{HardPrompt}$ | - | - | - | - | - | - | - | - | - | - | - | - |
| +PromptAlign$_{HardPrompt}$ | Acc. | ±.18 | ±.16 | ±.19 | ±.21 | ±.23 | ±.14 | ±.20 | ±.24 | ±.17 | ±.15 | ±.16 |
| | ECE | ±.19 | ±.14 | ±.22 | ±.18 | ±.17 | ±.12 | ±.15 | ±.20 | ±.19 | ±.13 | ±.17 |
| +PromptAlign$_{HardPrompt}$+C-TPT | Acc. | ±.19 | ±.15 | ±.18 | ±.12 | ±.14 | ±.22 | ±.19 | ±.21 | ±.22 | ±.16 | ±.14 |
| | ECE | ±.20 | ±.13 | ±.12 | ±.19 | ±.16 | ±.18 | ±.13 | ±.18 | ±.15 | ±.17 | ±.23 |

Table 15: **Standard Deviation of applying PromptAlign on Natural Distribution Shifts.**

| Method | | ImageNet-A | ImageNet-V2 | ImageNet-R | ImageNet-Sketch |
|---|---|---|---|---|---|
| CLIP-ViT-B/16$_{HardPrompt}$ | - | - | - | - | - |
| +PromptAlign$_{HardPrompt}$ | Acc. | ±.13 | ±.15 | ±.20 | ±.21 |
| | ECE | ±.18 | ±.22 | ±.11 | ±.24 |
| +PromptAlign$_{HardPrompt}$+C-TPT | Acc. | ±.10 | ±.15 | ±.19 | ±.19 |
| | ECE | ±.07 | ±.16 | ±.11 | ±.17 |

## A.12 APPLICABILITY OF C-TPT ON TRAINING-TIME PROMPT TUNING METHOD

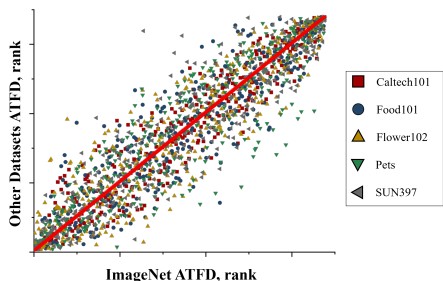

Figure 9: Plot illustrating the correlation between the ATFD rankings from ImageNet and those from various other datasets for 300 different hard prompts. A notable positive correlation is evident, as indicated by an **average Spearman rank correlation coefficient (Zar, 1972) of 0.92**.

Training-time prompt tuning methods like CoOp (Zhou et al., 2022a) and CoCoOp (Zhou et al., 2022b) typically involves training the prompt using a subset of ImageNet samples, emphasizing its generalizability across various datasets. This process often leads to a transition to datasets with differing class types. In this context, our focus turns to whether our proposed C-TPT regularization, which depends on the class types (as outlined in Equation 5), can apply to training time prompt tuning methods. The core of our evaluation in this section lies in determining whether the dispersion of text features among classes in a new test dataset can be generalized. For instance, **if a prompt shows high text feature dispersion in ImageNet, does it replicate this behavior in other datasets like Caltech, Pets, or Food101?**

We began with a detailed analysis of text feature dispersion using the prompts outlined in Appendix A.4, specifically focusing on the ImageNet classes. Our methodology involved ranking each prompt according to its text feature dispersion within the ImageNet classes. Subsequently, we extended to other datasets, ranking each prompt in the same manner based on its text feature dispersion in these other dataset classes. This process allowed us to directly compare each prompt's text feature dispersion ranking across different datasets with its initial ranking in ImageNet, providing a comprehensive view of their relative performance in diverse contexts. Our findings, as depicted in Figure 9, reveal a consistent trend: prompts that demonstrate high text feature dispersion within one dataset consistently exhibit a similar level of dispersion across other datasets.

This underlines the possibility of applying C-TPT regularization in training-time prompt tuning methods. By identifying prompts with high textual dispersion in one dataset, we can broadly apply this across diverse datasets. In this section, we perform the CoOp training with our proposed C-TPT on the cross-domain generalization setting proposed by Zhou et al. (2022a), which uses 16 samples per class of ImageNet to train a generalized prompt. We train on CLIP-RN50 with batch size 128, learning rate 0.002, and epoch 50 using SGD optimizer. The regularizer strength of C-TPT, which is denoted by $\lambda$, is set to 1. The result in Table 16 shows that C-TPT can work as a regularizer for better calibration error for the training-time prompt tuning method.

Table 16: **Fine-Grained Classification**. We report the **Acc.** ($\uparrow$) and **ECE** ($\downarrow$) of CoOp and CoOp + C-TPT on the CLIP-RN50. The values highlighted in **bold** signify the best ECE.

| Method | | ImageNet | Caltech | Pets | Cars | Flower102 | Food101 | Aircraft | SUN397 | DTD | EuroSAT | UCF101 | Average |
|---|---|---|---|---|---|---|---|---|---|---|---|---|---|
| CoOp | Acc. | 63.3 | 86.5 | 87.0 | 55.3 | 61.4 | 75.6 | 15.1 | 58.1 | 37.3 | 26.2 | 59.0 | 56.8 |
| | ATFD | 0.64 | 0.60 | 0.65 | 0.51 | 0.69 | 0.58 | 0.43 | 0.51 | 0.41 | 0.33 | 0.55 | 0.54 |
| | ECE | 3.00 | 3.32 | 6.75 | 7.27 | 5.01 | 3.06 | 7.20 | 3.59 | 12.4 | 20.2 | 5.02 | 6.98 |
| CoOp+C-TPT | Acc. | 62.4 | 87.9 | 87.0 | 55.3 | 61.8 | 74.6 | 14.0 | 59.2 | 36.3 | 25.3 | 58.2 | 56.5 |
| | ATFD | 0.92 | 0.91 | 0.66 | 0.74 | 0.75 | 0.67 | 0.60 | 0.64 | 0.65 | 0.56 | 0.71 | 0.71 |
| | ECE | **1.48** | **3.09** | **6.03** | **1.79** | **3.55** | **2.98** | **7.01** | **3.05** | **12.0** | **19.2** | **4.00** | **5.83** |

