# OpenReview forum: "C-TPT: Calibrated Test-Time Prompt Tuning for Vision-Language Models via Text Feature Dispersion"
_ICLR.cc/2024/Conference — ICLR 2024 poster_

### Official Review · Reviewer_CqA1 · 2023-10-23

**Soundness:** 3 good
**Presentation:** 3 good
**Contribution:** 3 good
**Rating:** 6
**Confidence:** 3

**Summary:**

This paper aims to calibrate the output of CLIP during test time. The authors discover the negative relationship between ATFD and ECE and use ATFD as regularization during test time prompt tuning.

**Strengths:**

1. The paper is well-written.
2. I appreciate the story of the paper.
3. I recognize the contribution of the paper --- calibration during test time. The proposed method is simple but effective. They maintain competitive performance with TPT and achieve better calibration.

**Weaknesses:**

Although I recognize the contribution of the paper, one practical problem is that the proposed method does not improve the metrics --- no positive influence is observed besides the ECE. A lower ECE does not show its significance in the tasks of the paper unlike the mentioned medical application. Hope authors can shed more light on this weaknesses

**Questions:**

One interesting thing is that CLIP (before TTA or fine-tuning) achieves the lowest ECE most time. It would be nice if the authors could shed light on this point.

---

> ### Author Response · Authors · 2023-11-15
>
> ## **[Response to Weakness]**
>
> Although the datasets used for the benchmark in our paper are not directly linked to safety-critical applications such as medical or automobile applications, the form of the task in that it is a classification task is identical. We used these benchmark datasets since they are the standard datasets used in CLIP prompt tuning literature.
>
> That being said, decreasing the ECE is essentially about reducing calibration error, ensuring that the model's confidence in its predictions accurately reflects the likelihood of those predictions being correct. Thus, having a low ECE is crucial in safety critical applications, such as medical applications, because it allows the use of the prediction confidence for decision-making. For example, **high-confidence predictions can be trusted, while low-confidence ones signal the need for human intervention or further analysis, thus preventing potential catastrophic outcomes.**
>
>
>
> ---
>
> ## **[Response to Question]**
>
> Thank you for highlighting the observation. The reasoning behind this is explained in **Section 4 - Observation 1**. At its core, the standard CLIP model is trained in an unsupervised manner, focusing on achieving a generalized text-image alignment without any specific objective of the classification task. Test-time prompt tuning (TPT) boosts the zero-shot classification accuracy of CLIP by reducing prediction entropy, which is in line with well-known test-time adaptation methods, such as TENT [1]. The reduction in entropy, reminiscent of classifiers trained with cross-entropy, leads the model to produce overconfident predictions [2]. Such overconfidence results in increased calibration errors (ECE).
>
> To further summarize, utilizing hard prompts in CLIP without any tuning does not involve entropy. However, the pursuit of increased classification accuracy during test time necessitates incorporating entropy minimization, which, unfortunately, worsens the calibration error. Our proposed method seeks to set a balance, aiming to alleviate this calibration degradation while simultaneously improving accuracy within the test-time prompt tuning scenario in CLIP. We have added a section **Appendix A.8 (Pareto Front: Visualizing the Effect of C-TPT)** to show that our method seeks the optimal balance between the accuracy and ECE by finding a solution in the Pareto Front. In the plot, we also show that the CLIP (before TTA or fine-tuning) does not lie in the Pareto Front and thus does not achieve the optimal balance between Accuracy and ECE; **although CLIP can have low ECE most of the time, it does not have optimal accuracy**.
>
>     [1] Wang et al., Tent: Fully Test-time Adaptation by Entropy Minimization, ICLR 2021
>     [2] Mukhoti et al., Calibrating Deep Neural Networks using Focal Loss, Neurips 2020

---

> ### Author Response · Authors · 2023-11-19
> **Further Updates on the Paper**
>
> We would like to inform you that **Appendix A.10 (Table 9, 10)** has been added to show the results of our method on different test-time prompt tuning method.
>
> Please let us know for further concerns. We are trying to address the reviewer's recommendations to the fullest extent possible. Your feedback is greatly appreciated. Thank you.

---

> > ### Author Response · Authors · 2023-11-22
> > **Reminder for Reviewer CqA1**
> >
> > Dear Reviewer CqA1,
> >
> >
> > Once again, we truly appreciate your time in reviewing our work. As the discussion period is finishing soon, we would be grateful if you could confirm whether our reply and modifications to the manuscript addressed your concerns.
> >
> >
> > Sincerely,
> >
> > Authors of Paper #5266

---

> > > ### Comment · Reviewer_CqA1 · 2023-11-23
> > > **Response**
> > >
> > > Thanks for the author's response. They address my concerns, so I remain my original score.

---

### Official Review · Reviewer_eM6j · 2023-10-28

**Soundness:** 3 good
**Presentation:** 4 excellent
**Contribution:** 3 good
**Rating:** 6
**Confidence:** 5

**Summary:**

This paper focused on the calibration issue in prompt tuning of vision-language models, especially for test-time prompt tuning, which is not fully explored in previous studies. The main findings are (1) test-time prompt tuning increases the calibration error, (2) the variance of calibration is high with different hard prompts, (3) calibration and text-feature dispersion are highly correlated. Based on the third finding, this work proposed a regularization term based on the text feature dispersion during test-time prompt tuning. Experiments on fine-grained classification and distribution shift datasets show that the proposed method achieves comparable performance with test-time prompt tuning method while lowering the calibration error.

**Strengths:**

+ The calibration error issue in test-time prompt tuning is not fully discussed in previous works. The research problem is novel and interesting.

+ The proposed regularization term is simple but effectively decreases the calibration error.

+ Experiments on fine-grained classification and distribution shift datasets show that the proposed method achieves comparable performance with test-time prompt tuning method while lowering the calibration error.

+ The paper is well written and organized. It is easy to follow the idea and figure out the main massages.

**Weaknesses:**

- Although the contributions of this work are obvious, it is somehow limited. The main messages I took from this paper are (1) this work pointed out the calibration error issue for test-time prompt tuning of VLMs, (2) provided an analysis on the relations between prompts and calibration error, (3) this work found the correlations between text-feature dispersion and calibration. Although the analysis is interesting and novel, it is not comprehensive enough. It would be interesting to know whether training-time prompt tuning also suffers from calibration error. If so, does the proposed method work for training-time prompt methods. If not, what is the reason. Also, the conclusions are empirical rather than theoretical. It is okay if the method is motivated merely on empirical observations, but the lack of theories decreased the contributions of this work.

- The proposed method is only adopted on TPT method. Do other test-time prompt tuning methods suffer from calibration error? Can the proposed method be applied to other methods? Only one method is discussed is not comprehensive to draw the conclusions.

- This work used 80 hard prompts to evaluate the impact of prompts on calibration error. However, according to the prompts shown in Appendix A.3, some of the prompts are not optimal. For example, "a black and white photo of a {class}" is not suitable to colored photos, and "a sketch of a {class}" may not work for photos. I am wondering whether the discrepancy between prompts and calibration error are due to the bad choices of prompts. If so, the conclusions about the relation between prompts and calibration error may not be convincing enough.

**Questions:**

Please see Weaknesses for detailed comments.

---

> ### Author Response · Authors · 2023-11-15
>
> ## **[Response to Weaknesses & Questions]**
>
> ### **Weakness 1**
>
> Training-time (i.e., Non-test time) prompt tuning methods such as CoOp and CoCoOp typically involve training the prompt with a subset of ImageNet samples and demonstrating the learned prompt's generalizability across various datasets. This process often involves transitioning to datasets with different class types. Since our proposed C-TPT regularization (Equation. 5) is dependent on the class types, such transitions inherently affect the dispersion of text features among the possible classes in the new test dataset.
>
> Given this context, our research has been specifically tailored to explore scenarios where there is a joint optimization of accuracy and calibration in the test-time adaptation where the possible classes are known.
>
> That being said, in **Appendix A.9 Table 8**, we have shown the results for TPT applied following the initial prompts set by learning-based **CoOp** and **CoCoOp**, which are a common settings shown in test-time prompt tuning literature. This demonstrates how test-time adaptation can benefit from the prior learning of non-test time prompt tuning methods, enhancing accuracy through improved prompt initialization. Our results indicate that while TPT can elevate the accuracy of CoOp and CoCoOp, it also impacts calibration. Our goal is to take advantage of TPT in enhancing accuracy while concurrently mitigating its effects on calibration, thereby achieving a more optimal balance for ECE.
>
> ---
>
> ### **Weakness 2**
>
> At the time of our submission, TPT was the only public test-time prompt tuning approach. We are currently conducting experiments with a recently published test-time prompt tuning method, DiffTPT [1], and will provide updates on these results. We thank the reviewer for the valuable suggestion!
>
> > [1] Feng at el., “Diverse Data Augmentation with Diffusions for Effective Test-time Prompt Tuning”, ICCV 2023
>
> ---
>
> ### **Weakness 3**
> We agree that certain prompts are better suited to specific datasets. However, our findings suggest that prompt appropriateness primarily influences the accuracy of the CLIP model rather than its calibration error (ECE).
>
> For illustration, consider the following examples from the ImageNet dataset using CLIP-RN50, categorized into *'appropriate'* and *'inappropriate'* prompts:
> * **Appropriate Prompts:**
>   + a photo of a - Acc: 58.2, ECE: 2.09, ATFD: 0.65
>   + a good photo of the - Acc: 58.1, ECE: 3.43, ATFD: 0.62
>   + a photo of the weird - Acc: 58.5, ECE: 5.35, ATFD: 0.57
>
> * **Inappropriate Prompts:**
>   + a sculpture of a - Acc: 53.2, ECE: 6.05, ATFD: 0.54
>   + graffiti of a - Acc: 53.8, ECE: 4.16, ATFD: 0.57
>   + a cartoon - Acc: 53.5, ECE: 2.21, ATFD: 0.59
>
> While *'appropriate prompts'* typically yield higher accuracy, the calibration error varies significantly within the group. Similarly *'inappropriate prompts'* tend to yield lower accuracy, and the calibration error varies significantly within the group. That is why in our analysis in Section 4 - Observation 2&3 and in Figure 2, we collated the prompts that yielded similar accuracy and tried to determine what caused the difference in the calibration error within the same accuracy group. **The examples provided show a negative correlation between ECE and ATFD within each group, which aligns with our paper's main findings.**

---

> > ### Comment · Reviewer_H39i · 2023-11-22
> >
> > Thanks for addressing my concerns. The added experiments on other test-time prompt tuning methods should have enhanced the credibility of this work. However, I still find it possible to apply the proposed method to training-time prompt tuning such as CoOp and CocoOp. The authors suggest that "Since our proposed C-TPT regularization (Equation. 5) is dependent on the class types, such transitions inherently affect the dispersion of text features among the possible classes in the new test dataset." Although the finetuning / transition will impact the dispersion of text features, the added regularization should also help with that. If indeed the proposed method does not work on training-time prompt tuning, it should be some other reasons. I believe that is worth clarified in this work, especially when training-time prompt tuning is more prominent in the research literature.

---

> > > ### Author Response · Authors · 2023-11-23
> > > **[Response for Reviewer H39i] Experimental Results on Training-time Prompt Tuning Method**
> > >
> > > Dear Reviewer H39i,
> > >
> > > Thank you for your constructive feedback. Upon reading your response, we agree that showing that a prompt trained to have high text feature dispersion on one dataset (e.g., ImageNet) can be generalized to different datasets will enhance the credibility of our work. As such, we have added section **Appendix A.12**, which incorporates an analysis of the generalizability of ATFD and experimental results on training-time prompt tuning method CoOp. We confirm in the experiment that C-TPT can work in training-time prompt tuning scenarios.
> > >
> > > While our paper primarily focuses on the application of test-time prompt tuning, the addition of **Appendix A.12** underscores the broader applicability of our approach in training-time contexts, thus enriching the overall scope and impact of our work.

---

> ### Author Response · Authors · 2023-11-19
> **Additional Response to Weakness 2: Experiments on other Test-time Prompt Tuning Methods**
>
> We thank the reviewer for the patience during this rebuttal period. We want to give an update on the experiments on other test-time prompt tuning methods.
> Specifically, we have applied C-TPT to DiffTPT [1] and PromptAlign [2], which are test-time prompt-tuning methods that have recently been published.
> * **DiffTPT [1]** is a method that adapts a similar method as TPT but leverages pre-trained diffusion models to generate diverse and informative augmented data.
> * **PromptAlign [2]** introduces a distribution alignment strategy for CLIP to improve test-time adaptation.
>
> >[1] Feng et al., “Diverse Data Augmentation with Diffusions for Effective Test-time Prompt Tuning”, ICCV 2023
>
> >[2] Hassan et al., "Align Your Prompts: Test-Time Prompting with Distribution Alignment for Zero-Shot Generalization", NeurIPS 2023
>
>
> \
> The following Table shows the results for **DiffTPT**. However, DiffTPT requires creating 64 different images per test sample for data augmentation. This process, unfortunately, takes a lot of time. For example, creating a complete test set for UCF101 (which is relatively small compared to other datasets) takes 1.5 days with 8 NVIDIA A100 80GB PCIe. Thus, for the DiffTPT method, we can only share the results for only 4 datasets during this rebuttal period, but the results indicate that our method can effectively reduce the ECE on DiffTPT as well.
>
> | Method              |                    | Car           | Flow                   | Euro          | UCF           | Avg.      |
> |-------------------------------------|---------------|---------------|----------------|---------------|---------------|---------------|
> |$\text{CLIP-ViTB/16}_{HardPrompt}$| Acc. (ECE)                  | 65.3 (4.25)  | 67.3 (3.00)  | 41.3 (7.40)   | 65.0 (3.59) | 59.7 (4.56)  |
> | +$\text{DiffTPT}_{HardPrompt}$| Acc. (ECE)                   |  67.0 (10.0)  | 69.9 (18.4)  | 43.1 (22.4)  | 67.7 (26.3) | 61.9 (19.3) |
> | +$\text{DiffTPT}_{HardPrompt}+\text{C-TPT}$| Acc. (ECE)               | 66.8 (**2.09**)  | 70.0 (**7.91**)  | 43.0  (**15.8**)  | 67.3 (**16.5**)| 61.8 (**10.6**)  |
>
> \
> Next, the following Table shows the results for **PromptAlign**. We show the results on both the fine-grained classification task and the natural data shift datasets. The results clearly show that C-TPT can effectively reduce the calibration error while taking advantage of the increased accuracy of the test-time tuning methods. This signifies that other test-time prompt tuning methods can benefit from C-TPT  in optimizing the prompt that achieves better calibration while increasing the accuracy.
>
> | Method       |              | Img           | Cal            | Pet           | Car           | Flow          | Food          | Air           | SUN           | DTD           | Euro          | UCF           | Avg.          |
> |-------------------------------------|--------------|---------------|----------------|---------------|---------------|---------------|---------------|---------------|---------------|---------------|---------------|---------------|---------------|
> | $\text{CLIP-ViTB/16}_{HardPrompt}$ | Acc. (ECE)           | 65.2 (3.65)  | 90.9 (7.51)   | 82.5 (2.91)  | 64.6 (2.49)  | 65.7 (4.70)  | 83.9 (2.78)  | 22.3 (7.09)  | 61.4 (3.33)  | 42.4 (4.94)   | 38.8 (13.4) | 64.8 (2.79)  | 62.0 (5.05)   |
> | +$\text{PromptAlign}_{HardPrompt}$| Acc. (ECE)                   | 72.1 (7.69)  | 94.1 (2.30)    | 90.5 (2.86)  | 68.0 (1.98)   | 72.1 (11.2) | 87.6 (3.04)  | 25.5 (8.30)  | 68.1 (8.36)  | 47.9 (25.6) | 44.8 (24.7) | 69.8 (12.1) | 67.3 (9.83)  |
> | +$\text{PromptAlign}_{HardPrompt} + \text{C-TPT}$| Acc. (ECE)             | 72.2 (**5.87**)  | 94.0 (**2.20**)  | 90.6 (**2.09**)  | 67.8 (**1.79**)   | 72.1 (**9.26**)  | 87.5 (**2.25**)  | 25.3 (**6.57**)  | 67.8 (**6.29**)  | 47.7 (**22.1**) | 45.9 (**21.8**) | 69.8 (**9.95**)  | 67.3 (**8.19**)   |
>
>
>
> | Method     |                         | ImageNet-A      | ImageNet-V2      | ImageNet-R       | ImageNet-Sketch  | Average         |
> |-------------------------------------|-----------------|-----------------|------------------|------------------|------------------|-----------------|
> | $\text{CLIP-ViTB/16}_{HardPrompt}$ | Acc. (ECE)                     | 47.8 (8.61)     | 60.8 (3.01)     | 74.0 (3.78)     | 46.1 (4.95)     | 57.2 (5.09)    |
> | +$\text{PromptAlign}_{HardPrompt}$ | Acc. (ECE)                       | 55.8 (16.1)    | 65.3 (10.8)    | 78.5 (4.70)     | 50.3 (16.8)     | 62.5 (12.1)   |
> | +$\text{PromptAlign}_\text{HardPrompt}+ \text{C-TPT}$ | Acc. (ECE)                 | 55.6 (**13.1**)    | 65.1 (**7.10**)     | 78.6 (**2.98**)     | 50.2 (**12.8**)    | 62.4 (**9.00**)    |
>
> \
> We have included these results in **Appendix A.10 (Table 6 and Table 7)**. We thank the reviewer again for the valuable suggestions. If there are any further concerns, please let us know. We would like to accommodate the reviewer's suggestions as much as possible.

---

> > ### Author Response · Authors · 2023-11-22
> > **Reminder for Reviewer eM6j**
> >
> > Dear Reviewer eM6j,
> >
> > \
> > Once again, we truly appreciate your time in reviewing our work. As the discussion period is finishing soon, we would be grateful if you could confirm whether our reply and modifications to the manuscript addressed your concerns.
> >
> >
> > \
> > Sincerely,
> >
> > Authors of Paper #5266

---

> > > ### Comment · Reviewer_eM6j · 2023-11-22
> > >
> > > Thank the authors for the detailed responses and updated results. The additional results on training-time prompting methods and test-time prompting methods addressed my concerns. However, my concern about the lack of theoretical evidence is not addressed. Considering the main messages I learned from this work, I think this paper is a borderline one. I will join the reviewer-AC discussion and further discuss the novelty with other colleagues.

---

> ### Author Response · Authors · 2023-11-23
> **[Response for Reviewer eM6j]**
>
> Dear Reviewer eM6j,
>
> Thank you for your response. We acknowledge the concerns raised on theoretical evidence and appreciate the constructive feedback. However, we believe that our methodology was systematically driven and hope the findings will benefit the broader community. Our methodology represents a pioneering effort in the field, being the first to achieve calibration in CLIP prompt tuning scenarios without the need for labeled data.
>
> Nonetheless, we truly appreciate the time and effort of the reviewer on providing valuable feedbacks. Thank you!

---

### Official Review · Reviewer_H39i · 2023-10-30

**Soundness:** 3 good
**Presentation:** 3 good
**Contribution:** 2 fair
**Rating:** 6
**Confidence:** 4

**Summary:**

This paper first makes observations on the effect of text prompt to the CLIP model calibration (consistency of predicted confidence with the real accuracy), where there is a negative correlation between the dispersion of the text features and the calibration error. Then it proposes the Average Text Feature Dispersion (ATFD) to measure this dispersion of the text features. It then proposes a loss function based on the ATFD to increase the ATFD thus improving calibration during test-time adaptation. The approach is validated on different datasets with CLIP models and shows the improved results with smaller calibration error and similar accuracy compared to an existing test-time prompt tuning methods.

**Strengths:**

-	The authors make several observations before introduction their technique to improve the calibration property of the model. The observations (figure 1&2) are intuitive and clearly shows the correlation between the calibration error and the ATFD metric.
-	The proposed ATFD loss combined with an existing test-time adaptation method shows consistently improved results on various downstream tasks (ImageNet, Caltech, Pets etc)
-	Paper writing is clear and easy to understand. The key claims are well supported by the evidence from the experiments.

**Weaknesses:**

-	The application of improving calibration on just one algorithm (test-time prompt tuning) is kind of limited. I would expect this method to help improve different prompt tuning methods such as CoOp. Additional experiments on other prompt tuning tasks are strongly recommended to improve the impact of this work.
-	Figure 5 is problematic. First, the color scheme of Figure 5 is incorrect. The legends of (a) and (b) both have the black color, but the dots in the prompt visualization figure does not have the black dots. And the class visualization is weird. Do the authors want to show the class of some samples on the visualized plan? Then the location of the class samples should be different from the class prompt. However, I find that in Figure 5(a) the class samples are perfectly aligned with the class prompts.

**Questions:**

Is there specific reason to prevent this method to be used for other prompt tuning methods such as CoOp and CoCoOp (Conditional Prompt Learning for Vision-Language Models) ?

---

> ### Author Response · Authors · 2023-11-15
>
> ## **[Response to Weaknesses & Questions]**
>
> ### **Weakness 1**
>
> * Non-test time prompt tuning methods such as CoOp and CoCoOp typically involve training the prompt with a subset of ImageNet samples, with an emphasis on demonstrating the prompt's generalizability across various datasets. This process often involves a transition to datasets with different class types. Since our proposed C-TPT regularization (Equation. 5) is dependent on the class types, such transitions inherently affect the dispersion of text features among the possible classes in the new test dataset.
> \
> \
> Given this context, our research has been specifically tailored to explore scenarios where there is a joint optimization of accuracy and calibration in the test-time adaptation where the possible classes are known.
> \
> \
> That being said, in **Appendix A.9 Table 8**, we have shown the results for TPT applied following the initial prompts set by learning-based **CoOp** and **CoCoOp**, which are a common settings shown in test-time prompt tuning literature. This demonstrates how test-time adaptation can benefit from the prior learning of non-test time prompt tuning methods, enhancing accuracy through improved prompt initialization. Our results indicate that while TPT can elevate the accuracy of CoOp and CoCoOp, it also impacts calibration. Our goal is to take advantage of TPT in enhancing accuracy while concurrently mitigating its effects on calibration, thereby achieving a more optimal balance for ECE.
> #
> * Furthermore, we are currently conducting experiments with a recently proposed test-time prompt tuning method, DiffTPT [1], and will provide updates on these results. We thank the reviewer for the constructive input.
>
> > [1] Feng at el., “Diverse Data Augmentation with Diffusions for Effective Test-time Prompt Tuning”, ICCV 2023
>
> ---
>
> ### **Weakness 2**
>
> * Thank you for pointing out the concerns regarding Figure 5. The black dots are present in the plot (which can be slightly seen when zoomed in) but may not be immediately visible due to overlapping with other points.  **Thus, we have revised Figure 5 to improve the visibility of these overlapped points.**
> #
> * The plot in the Prompt Visualization and the Class Visualization should be the same. In Prompt Visualization, each point represents a text feature for all the possible classes in the Caltech101 dataset, with each specific prompt indicated by a unique color as per the legend. In the Class Visualization, we have only changed the coloring scheme so that the same class has the same color. This was to illustrate that well-calibrated prompts tend to locate the class samples in the same place with one another, while the bad-calibrated prompts tend to cluster all the possible classes together.
>
> Thank you for letting us know about the visibility of the figure, which could potentially confuse readers. Please let us know if you have any further questions or require additional clarification.

---

> ### Author Response · Authors · 2023-11-19
> **Additional Experimental Results on Other Test-time Prompt Tuning Methods**
>
> We thank the reviewer for the patience during this rebuttal period. We want to give an update on the experiments on other test-time prompt tuning methods.
> Specifically, we have applied C-TPT to DiffTPT [1] and PromptAlign [2], which are test-time prompt-tuning methods that have recently been published.
> * **DiffTPT [1]** is a method that adapts a similar method as TPT but leverages pre-trained diffusion models to generate diverse and informative augmented data.
> * **PromptAlign [2]** introduces a distribution alignment strategy for CLIP to improve test-time adaptation.
>
> >[1] Feng et al., “Diverse Data Augmentation with Diffusions for Effective Test-time Prompt Tuning”, ICCV 2023
>
> >[2] Hassan et al., "Align Your Prompts: Test-Time Prompting with Distribution Alignment for Zero-Shot Generalization", NeurIPS 2023
>
> \
> The following Table shows the results for **DiffTPT**. However, DiffTPT requires creating 64 different images per test sample for data augmentation. This process, unfortunately, takes a lot of time. For example, creating a complete test set for UCF101 (which is relatively small compared to other datasets) takes 1.5 days with 8 NVIDIA A100 80GB PCIe. Thus, for the DiffTPT method, we can only share the results for only 4 datasets during this rebuttal period, but the results indicate that our method can effectively reduce the ECE on DiffTPT as well.
>
> | Method              |                    | Car           | Flow                   | Euro          | UCF           | Avg.      |
> |-------------------------------------|---------------|---------------|----------------|---------------|---------------|---------------|
> |$\text{CLIP-ViTB/16}_{HardPrompt}$| Acc. (ECE)                  | 65.3 (4.25)  | 67.3 (3.00)  | 41.3 (7.40)   | 65.0 (3.59) | 59.7 (4.56)  |
> | +$\text{DiffTPT}_{HardPrompt}$| Acc. (ECE)                   |  67.0 (10.0)  | 69.9 (18.4)  | 43.1 (22.4)  | 67.7 (26.3) | 61.9 (19.3) |
> | +$\text{DiffTPT}_{HardPrompt}+\text{C-TPT}$| Acc. (ECE)               | 66.8 (**2.09**)  | 70.0 (**7.91**)  | 43.0  (**15.8**)  | 67.3 (**16.5**)| 61.8 (**10.6**)  |
>
> \
> Next, the following Table shows the results for **PromptAlign**. We show the results on both the fine-grained classification task and the natural data shift datasets. The results clearly show that C-TPT can effectively reduce the calibration error while taking advantage of the increased accuracy of the test-time tuning methods. This signifies that other test-time prompt tuning methods can benefit from C-TPT in optimizing the prompt that achieves better calibration while increasing the accuracy.
>
> | Method       |              | Img           | Cal            | Pet           | Car           | Flow          | Food          | Air           | SUN           | DTD           | Euro          | UCF           | Avg.          |
> |-------------------------------------|--------------|---------------|----------------|---------------|---------------|---------------|---------------|---------------|---------------|---------------|---------------|---------------|---------------|
> | $\text{CLIP-ViTB/16}_{HardPrompt}$ | Acc. (ECE)           | 65.2 (3.65)  | 90.9 (7.51)   | 82.5 (2.91)  | 64.6 (2.49)  | 65.7 (4.70)  | 83.9 (2.78)  | 22.3 (7.09)  | 61.4 (3.33)  | 42.4 (4.94)   | 38.8 (13.4) | 64.8 (2.79)  | 62.0 (5.05)   |
> | +$\text{PromptAlign}_{HardPrompt}$| Acc. (ECE)                   | 72.1 (7.69)  | 94.1 (2.30)    | 90.5 (2.86)  | 68.0 (1.98)   | 72.1 (11.2) | 87.6 (3.04)  | 25.5 (8.30)  | 68.1 (8.36)  | 47.9 (25.6) | 44.8 (24.7) | 69.8 (12.1) | 67.3 (9.83)  |
> | +$\text{PromptAlign}_{HardPrompt} + \text{C-TPT}$| Acc. (ECE)             | 72.2 (**5.87**)  | 94.0 (**2.20**)  | 90.6 (**2.09**)  | 67.8 (**1.79**)   | 72.1 (**9.26**)  | 87.5 (**2.25**)  | 25.3 (**6.57**)  | 67.8 (**6.29**)  | 47.7 (**22.1**) | 45.9 (**21.8**) | 69.8 (**9.95**)  | 67.3 (**8.19**)   |
>
>
>
> | Method     |                         | ImageNet-A      | ImageNet-V2      | ImageNet-R       | ImageNet-Sketch  | Average         |
> |-------------------------------------|-----------------|-----------------|------------------|------------------|------------------|-----------------|
> | $\text{CLIP-ViTB/16}_{HardPrompt}$ | Acc. (ECE)                     | 47.8 (8.61)     | 60.8 (3.01)     | 74.0 (3.78)     | 46.1 (4.95)     | 57.2 (5.09)    |
> | +$\text{PromptAlign}_{HardPrompt}$ | Acc. (ECE)                       | 55.8 (16.1)    | 65.3 (10.8)    | 78.5 (4.70)     | 50.3 (16.8)     | 62.5 (12.1)   |
> | +$\text{PromptAlign}_\text{HardPrompt}+ \text{C-TPT}$ | Acc. (ECE)                 | 55.6 (**13.1**)    | 65.1 (**7.10**)     | 78.6 (**2.98**)     | 50.2 (**12.8**)    | 62.4 (**9.00**)    |
>
> \
> We have included these results in **Appendix A.10 (Table 6 and Table 7)**. We thank the reviewer again for the valuable suggestions. If there are any further concerns, please let us know. We would like to accommodate the reviewer's suggestions as much as possible.

---

> ### Author Response · Authors · 2023-11-22
> **Reminder for Reviewer H39i**
>
> Dear Reviewer H39i,
>
> Once again, we truly appreciate your time in reviewing our work. As the discussion period is finishing soon, we would be grateful if you could confirm whether our reply and modifications to the manuscript addressed your concerns.
>
> Sincerely,
>
> Authors of Paper #5266

---

> > ### Author Response · Authors · 2023-11-23
> > **[Response for Reviewer H39i] Experimental Results on Training-time Prompt Tuning Method**
> >
> > Dear Reviewer H39i,
> >
> > We have noticed that your previous response has been posted under Reviewer eM6j's section. We have further replied to your feedback on that section. We would greatly appreciate if you could take a look at the comment and the additional experimental results we have included on the training-time prompt tuning method (**Appendix A.12**). We are grateful for your time and hope we have addressed your concerns. Thank you!

---

### Official Review · Reviewer_fxyk · 2023-10-30

**Soundness:** 3 good
**Presentation:** 4 excellent
**Contribution:** 2 fair
**Rating:** 6
**Confidence:** 4

**Summary:**

This paper studies an interesting problem related to test-time prompt tuning of a vision-language model. Through a series of observations, this paper finds out that the calibration is linked to the choice of prompts. Therefore, this paper attempts to improve the calibration error (ECE) by using the average dispersion of text features. The main idea is to distribute the text features in the feature representation space. The experiments show that the proposed method successfully improves the ECE of test-time prompt methods. The topic of this paper is relatively interesting and proposes a novel perspective for prompt tuning.

**Strengths:**

1. The solution (i.e., ATFD) of this paper is reasonable. The choice of prompts affects the ECE metric, indicating that the feature representation is well-calibrated. Therefore, prompt tuning for placing textual class vectors separately is important.
2. The experiments of this paper are comprehensive, and the improvement of ECE is significant. Both the ablation study and additional analysis can provide valuable information to readers.
3. The presentation of this paper is good, and the technical content is easy to follow.

**Weaknesses:**

1. The experimental results show that the accuracy (ACC) of C-TPT drops in many cases. However, the ECE and ACC are a trade-off in most cases. The author should explain whether the drop in accuracy is worth the improvement of ECE. My advice is that authors can provide a Pareto frontier of ECE vs. ACC for C-TPT methods and hand-crafted prompts to demonstrate that C-TPT achieves a superior solution, rather than simply obtaining a prompt with higher ECE and lower ACC.
2. The related work should discuss a related field, i.e., Test-time Adaptation [1-5].

**Reference:**

[1] Shuaicheng Niu, Jiaxiang Wu, Yifan Zhang, Yaofo Chen, Shijian Zheng, Peilin Zhao, Mingkui Tan: Efficient Test-Time Model Adaptation without Forgetting. ICML 2022: 16888-16905

[2] Hao Zhao, Yuejiang Liu, Alexandre Alahi, Tao Lin: On Pitfalls of Test-Time Adaptation. ICML 2023

[3] Zhi Zhou, Lan-Zhe Guo, Lin-Han Jia, Dingchu Zhang, Yu-Feng Li: ODS: Test-Time Adaptation in the Presence of Open-World Data Shift. ICML 2023

[4] Tong Wu, Feiran Jia, Xiangyu Qi, Jiachen T. Wang, Vikash Sehwag, Saeed Mahloujifar, Prateek Mittal: Uncovering Adversarial Risks of Test-Time Adaptation. ICML 2023

[5] Dequan Wang, Evan Shelhamer, Shaoteng Liu, Bruno A. Olshausen, Trevor Darrell: Tent: Fully Test-Time Adaptation by Entropy Minimization. ICLR 2021

**Questions:**

Please refer to Weakness 1 & 2. I will raise my score if you address my concerns.

[UPDATE]
I have raised my score.

**Details Of Ethics Concerns:**

No ethics concerns

---

> ### Author Response · Authors · 2023-11-15
>
> ## **[Response to Weaknesses & Questions]**
>
> ### **Weakness 1**
>
> We appreciate your insightful feedback regarding the trade-off between accuracy and calibration error (ECE) observed in our C-TPT method. In response to your suggestion, demonstrating this trade-off through a Pareto front would be a valuable addition to our analysis.
>
> In the revised manuscript, we have expanded **Appendix A.8** to include a Pareto front analysis of ECE versus Acc for both C-TPT and hand-crafted prompts. **Figure 8** illustrates the trade-offs involved and shows that our C-TPT method does not merely achieve higher ECE at the expense of lower Acc. Instead, it seeks an optimal balance between the two.
>
> We believe that this addition strengthens the findings and overall contributions of our paper by providing a more detailed picture of the performance characteristics of C-TPT. Thank you again for the valuable insight.
>
> ---
> ### **Weakness 2**
>
> Thank you for highlighting the importance of discussing test-time adaptation in greater detail. While we have referenced some relevant methods in Section 1 (Introduction) and Section 4 (Revisiting the Calibration of CLIP Models - Observation 1), we agree that test-time adaptation methods should have a separate related work section. We have included the relevant works in **Section 2 (Related Works)** of the revised manuscript.

---

> ### Comment · Reviewer_fxyk · 2023-11-15
>
> Thank you for responding to my concerns and addressing them effectively. As a result, I have increased my score.

---

> > ### Author Response · Authors · 2023-11-15
> >
> > Thank you again for acknowledging the contribution our work and giving the opportunity to strengthen the paper!

---

> > > ### Author Response · Authors · 2023-11-19
> > > **Further Updates on the Paper**
> > >
> > > We would like to inform you that **Appendix A.10 (Table 9, 10)** has been added to show the results of our method on different test-time prompt tuning method.
> > >
> > > If there are any further concerns, please let us know. We would like to accommodate the reviewer's suggestions as much as possible. Thank you.

---

### Author Response · Authors · 2023-11-15
**General Response to Reviewers**

We would like to thank the reviewers for the detailed review and feedback. We have addressed each noted question separately. We have uploaded a revised version of the manuscript with the following additions (revised parts are indicated in red):

* **Appendix A.8** has been expanded to include the visualization of the effect of C-TPT to show that C-TPT provides a solution in the Pareto front.

* Test-time adaptation section has been added to the **Section 2: Related Works**.

* **Figure 5** has been modified for better visualization.

Moreover, we are currently running experiments to show the effectiveness of our method on the recently published test-time prompt tuning method (DiffTPT [1]). We will provide updates on this result once it is available. Thank you.

> [1] Feng at el., “Diverse Data Augmentation with Diffusions for Effective Test-time Prompt Tuning”, ICCV 2023

---

> ### Author Response · Authors · 2023-11-19
> **[Update] Manuscript 2nd Revision Uploaded: Addition of different test-time prompt tuning method**
>
> We thank the reviewers again for the valuable suggestions and patience during the rebuttal period. We have just uploaded a revised version with the following additional change:
>
> * **Appendix A.10 (Table 9, 10)** has been added to show the results of our method on different test-time prompt tuning method.
>
> We would greatly appreciate if the reviewers read through the revised manuscript. Please let us know for further concerns. Thank you.

---

> ### Author Response · Authors · 2023-11-23
> **[Update] Manuscript 3rd Revision Uploaded: Addition of training-time prompt tuning method**
>
> We have just uploaded a revised version with the following additional change:
>
> * **Appendix A.12** has been added to show the applicability of C-TPT on training-time prompt tuning method.
>
> We thank all the reviewers for their valuable feedback on the work.
>
> \
> Sincerely,
>
> Authors of Paper #5266

---

### Meta-Review · Area_Chair_3hGE · 2023-12-02

**Metareview:**

This paper makes an interesting observations for text prompt on the CLIP model calibration and proposes the Average Text Feature Dispersion (ATFD) to measure this dispersion of the text features. A loss function is then proposed to improve calibration during test-time adaptation. Results on different datasets show the effectiveness of the proposed method.

**Strengths**


- Several interesting observations are provided to guide the design of calibration model.

- The proposed ATFD loss shows consistently improvements on various datasets.

- Paper writing is clear and easy to understand. The key claims are well supported by the evidence from the experiments.

**Weaknesses**

Most of the concerns are addressed during rebuttal. However, several concerns still remain.

- The significance of the method can be further justified, which is not very clear.

- The proposed method is lack of theoretical evidence, which is a pity for ICLR.

- More discussions with related works should be comprehensively provided in the final version.

**Justification For Why Not Higher Score:**

The significance of the method is not well justified and the proposed method is lack of theoretical evidence.

**Justification For Why Not Lower Score:**

This paper provides interesting observations for CLIP model calibration as well as proposes a tailor-made approach which achieves consistent improvements on various datasets. This work may be beneficial for someone who cares about the reliability of the network.

---

### Decision · Program_Chairs · 2024-01-16

Accept (poster)